# Configurations and Applications of Multi-Agent Hybrid Drone/Unmanned Ground Vehicle for Underground Environments: A Review

Chris Dinelli [1], John Racette [1], Mario Escarcega [1], Simon Lotero [2], Jeffrey Gordon [2], James Montoya [1], Chase Dunaway [1], Vasileios Androulakis [2], Hassan Khaniani [3], Sihua Shao [4], Pedram Roghanchi [2] and Mostafa Hassanalian [1,*]

1   Department of Mechanical Engineering, New Mexico Tech, Socorro, NM 87801, USA
2   Department of Mineral Engineering, New Mexico Tech, Socorro, NM 87801, USA
3   Petroleum Recovery Research Center, New Mexico Tech, Socorro, NM 87801, USA
4   Department of Electrical Engineering, New Mexico Tech, Socorro, NM 87801, USA
*   Correspondence: mostafa.hassanalian@nmt.edu

**Abstract:** Subterranean openings, including mines, present a unique and challenging environment for robots and autonomous exploration systems. Autonomous robots that are created today will be deployed in harsh and unexplored landscapes that humanity is increasingly encountering in its scientific and technological endeavors. Terrestrial and extraterrestrial environments pose significant challenges for both humans and robots: they are inhospitable and inaccessible to humans due to a lack of space or oxygen, poor or no illumination, unpredictable terrain, a GPS-denied environment, and a lack of satellite imagery or mapping information of any type. Underground mines provide a good physical simulation for these types of environments, and thus, can be useful for testing and developing highly sought-after autonomous navigation frameworks for autonomous agents. This review presents a collective study of robotic systems, both of individual and hybrid types, intended for deployment in such environments. The prevalent configurations, practices for their construction and the hardware equipment of existing multi-agent hybrid robotic systems will be discussed. It aims to provide a supplementary tool for defining the state of the art of coupled Unmanned Ground Vehicle (UGV)–Unmanned Aerial Vehicle (UAV) systems implemented for underground exploration and navigation purposes, as well as to provide some suggestions for multi-agent robotic system solutions, and ultimately, to support the development of a semi-autonomous hybrid UGV–UAV system to assist with mine emergency responses.

**Keywords:** subterranean autonomous exploration; drones; bio-inspired robots; coupled UGV–UAV systems; multi-agent hybrid robotic systems

## 1. Introduction

The ever-increasing demands of society for material resources are leading scientific and industrial societies to explore and exploit harsh environments. Terrestrial and extraterrestrial environments pose significant challenges for both humans and robots: they are inhospitable and inaccessible to humans due to a lack of space or oxygen, poor or no illumination, unpredictable terrain, a GPS-denied nature (Global Positioning System), and a lack of satellite imagery or mapping information of any type [1–4]. Underground mines and openings are a good physical simulation of these types of environments, and thus, can be useful for testing and developing highly sought-after autonomous navigation frameworks for robots [5,6]. A variety of dynamic and static obstacles, such as machinery and heavy equipment, infrastructure, mine workers, falling material, and suspending dust, prevalent in mines during normal mining operations are additional complexity factors [7]. Furthermore, during mining emergency situations, catastrophes can create even more complex

obstacles and compounding dangers in these environments, which, to name a few, are: flash fires, massive shaft or roof collapses, water, a shortage of oxygen, wind, and water. Thus, mining emergencies can worsen the hazardous conditions for humans inside the mine, as well as block the escape paths and completely trap them.

In such emergencies, specialized first responders are deployed to rescue the trapped miners, so long as the mine is safe enough for the team to enter and effectively execute their mission. These missions are not always successful, and even upon the rescue of the trapped mine workers, a fully successful outcome is not guaranteed. Sending these search and rescue teams inside a mine during these emergencies requires the first responders to risk their lives and well-being. Requiring human responders to voluntarily enter an inhospitable, dynamically changing, hazardous, and potentially deadly environment with no guarantee of a safe return is an inefficient solution.

Naturally, integrating the recent scientific and technological advances in such missions can play a critical role in increasing the chances for successful outcomes and saving many lives [5–7]. Equipping these search and rescue teams with advanced tools and frameworks by harnessing the rapid technological advances in robotics, computer hardware, and artificial intelligence is the most sensible next step in the creation of future mines. The use of robots for improving safety and efficiency in underground spaces is not a new concept, although a new tool is at the forefront of this technology. Autonomous, unmanned robotic agents are independent robotic elements that are capable of user-independent operation and problem solving, hence rendering human supervision obsolete. Removing the human operator from the loop means that these robots can perform missions in environments that are either inaccessible or unsafe for human agents. Furthermore, while a single autonomous agent can accomplish many tasks independently, there are some constraints that necessitate a combination of agents working in tandem. In some cases, the combination of multiple robotic agents can dramatically increase the system's overall efficiency by eliminating the risk of a single-point failure, which stems from the complete reliance on a single independent agent.

Conceptually, these autonomous and semi-autonomous multi-agent systems have as much variation and as wide a range of applications as those in the modern mechatronics and robotics fields do. The advance of collaborative swarms of UGVs and drones that are able to operate, navigate, communicate, and even install and assemble "each other" as a symbiotic team will play a critical role in the industry, science, and society in the future [8]. Highly durable and technologically robust multi-agent hybrid robotic systems (MAHRSs) could provide solutions not only to modern problems and mysteries, but also to ones that humanity has yet to encounter. Multi-agent systems that require little to no human input are a future solution to many of the infrastructure issues that modern humans face today. Specifically, when they are applied to subterranean and other GPS-denied environments, these systems can provide an inherently modular and flexible solution to the dynamic and unpredictable obstacles that are present. The application of these systems to modern problems will have long-standing effects on the safety, flexibility, and efficiency of accessing some of the harshest environments on our planet and far beyond its limits.

The MAHRSs discussed in this review are deployed exclusively in GPS-denied environments with applications ranging from mine safety to archaeology and excavation to structural health monitoring. As the design of these robotic systems becomes more robust in terms of intrinsic safety, mission efficiency, and operational time optimization, future terrestrial and extraterrestrial exploration missions will grant us access to the landscapes and terrains that humans cannot currently reach. To understand the future applications of these systems, one must first examine how science and industry are shaping the landscape of robotics in underground spaces today.

This review focuses primarily on unmanned ground and aerial vehicles, either as stand-alone agents or as multi-agent systems (hybrid) in underground spaces, and especially, fully GPS-denied ones. The principal motivation for this study stems from the desire to improve the safety and health of mine rescue teams deployed during mine emergencies.

The impact of smart, multi-agent systems on mitigating the exposure of human rescuers to the occupational hazards prevalent in a mine during an emergency can potentially be enormous. The same smart systems can be deployed in similarly harsh environments, terrestrial or extraterrestrial ones, which are inaccessible or inhospitable to humans. Therefore, the scope of this study is to provide a comprehensive review of the design elements and the operation constraints of existing robotic systems. To understand the applications of autonomous robots in these environments, one must evaluate the applications of their mechanical predecessors, the present state-of-the-art ones, and the envisioned future applications. Hence, this study examines UGVs and drones that have already been designed and deployed in underground spaces. This means that some aspect their configuration (size, shape, locomotion type, etc.) has been optimized for these environments. This review also discusses bio-inspired robots that have not, but could potentially, be optimized for auxiliary individual or swarm operations. The ultimate scope of this review is to serve as a collection of literature reviews, Mineral Engineering R&D, and Intelligent Robotic Systems R&D, which aim to improve and expedite future R&D on intelligent underground systems, further understand the impact of the technological advancements upon human safety and work efficiency in harsh terrestrial and extraterrestrial environments, and finally, to inspire designers and innovators of such systems.

The rest of the paper is organized as follows: Section 2 discusses extensively the fundamental classification of unmanned systems, with the emphasis given to Unmanned Ground Vehicles (UGVs) and Unmanned Aerial Vehicles (UAVs), as well as hybrid UAV–UGV (single-agent systems). Section 3 focuses on cooperative UGV–UAV systems (multi-agent systems). In Section 4, a discussion of mines and underground hazards and challenges stemming from the nature of these environments is presented, while Section 5 provides detailed underground applications of hybrid multi-agent systems. Finally, Section 6 discusses the main takeaway from this review and presents some conclusions.

## 2. Classification of Unmanned Systems

As considered in this review, MAHRSs can be described as the coupling of two or more robotic systems to complete missions as a team. This concept can be compared to biological symbiosis, defined as "any type of a close and long-term biological interaction between two different biological organisms, be it mutualistic, commensalistic, or parasitic" [9]. While the roles of each robot within the system vary in complexity and importance, the overarching objective of using a multi-part hybrid robotic system is to utilize the specializations of many robotic designs, instead of creating one robot that can accomplish every task as an individual. Therefore, each robot is designed to accomplish a mission-specific objective that is part of a bigger objective. Hence, the term MAHRS encompasses not only the design and function of the individual robotic agents, but also the interfacing, connectivity, and collaboration rules between the robotic agents and the simultaneous (passive/active) operation of these multiple specialized robotic agents in order to design a holistically efficient system. It is vital to utilize this design concept to understand the state-of-the-art UGVs and drones as they can be found in commercial marketplaces, scientific publications, industrial settings, and military applications. To begin with, one must first become familiar with the fundamental classification of the different robots, the nomenclature that exists to define these systems, and the individual functionality of each classification. In the following subsections, single robot systems are reviewed, and specifically UGVs, UAVs (or drones), and single hybrid UGV–UAV robots (see Figure 1).

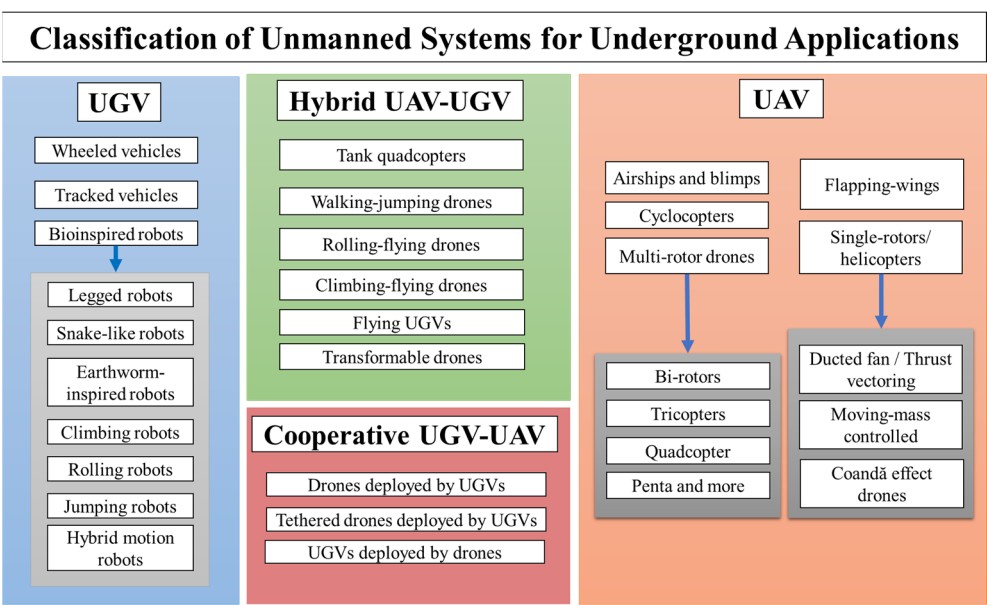

**Figure 1.** Classification of unmanned systems for underground applications.

## 2.1. Unmanned Ground Vehicles (UGV)

Multiple configurations of unmanned ground vehicles have been developed and implemented for various purposes. In this subsection, a review of UGVs is presented with emphasis on wheeled, tracked, as well as bio-inspired robots. In the latest category, a brief review of legged, snake-like, earthworm-inspired, climbing, rolling, jumping, and hybrid motion robots is discussed.

### 2.1.1. Wheeled Vehicles

Wheeled UGVs are one of the most common types of terrestrial robots deployed for environment exploration, whether it is a well-structured environment or an unknown, harsh environment [10]. Commonly, a UGV is designed to execute a specialized mission based on the output of its exploration and mapping module (see Figure 2). Wheel-based designs come in a wide range of configurations, from the widely used four-wheeled designs to highly complex multi-wheeled rovers designed to traverse highly uneven landscapes [11]. The size, design, and type of the wheels can vary based on the working space and the specialized missions assigned to the UGV. Four-wheeled designs, which are mechanically simpler, enable efficient locomotion on relatively even surfaces. This is the reason that the four-wheeled configuration prevails in most passenger vehicles that are in operation today. By applying torque to the center of each wheel, the robots can move forwards and backward, while by manipulating the steering angle of the wheels or their adjoining shafts, the robots can turn, spin, flip, or perform similar operations. Autonomous sensing and perception capabilities in modern robotics mean that these systems, such as most of the systems discussed in this review, can be autonomously operated by defining a small number of inputs.

Wheeled robots have been used in many scientific fields of research, industrial purposes, and military applications [12]. International manufacturers and distributors make various commercial platforms that are available in the marketplace. These platforms can be equipped with a wide variety of sensors, mechanical arms and handlers, and artificial intelligence to aid processes in nearly every industry and application. However, implementing a terrestrial robot to solve a given problem requires researchers or inventors to select between constructing a customized robot or utilizing an existing commercial robot. The choice must be made upon careful consideration of the intended implementation and the specialized missions that will be assigned to the robot. The literature is abundant on

such specialized robots, with applications ranging from agricultural monitoring and pruning (see Figure 2a) to extraterrestrial exploration (see Figure 2b).

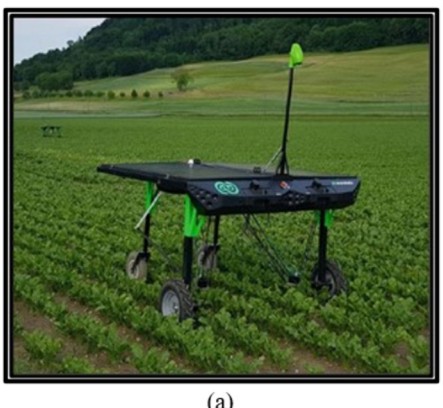 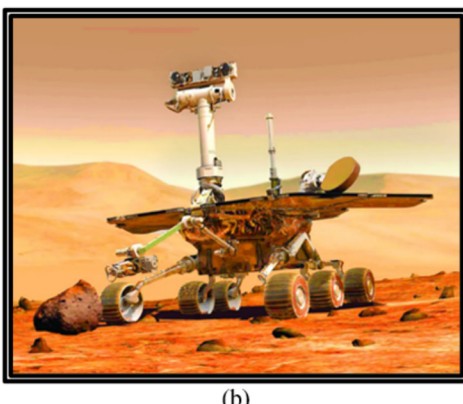

(a)    (b)

**Figure 2.** Examples of wheeled UGVs with specialized functionalities: (**a**) four-wheeled UGV drives through seedlings, picking weeds between rows [13]; (**b**) six-wheeled UCV traverses Mars' surface.

The movement of a wheeled robot, and thus its capabilities in each working space, is determined by the number and configuration of the wheels and the wheel axles. For example, a two-wheeled robot with differential drive (see Figure 3a) has three degrees of freedom (DOF), i.e., the position (x, y) and the heading ($\varphi$) and two actuators, i.e., the motors on the two wheels [13]. These motors enable the robot to move forwards and backward and rotate in place, but the robot cannot move laterally without first rotating. In comparison, a four-wheeled robot with Ackermann steering has the same three DOF, but two different actuators, i.e., a motor and a steering wheel (see Figure 3b). The motor enables the robot to move forwards and backward, while the steering wheel can only orient the former movement. As a result, the four-wheeled robot can neither rotate in place nor move laterally [13].

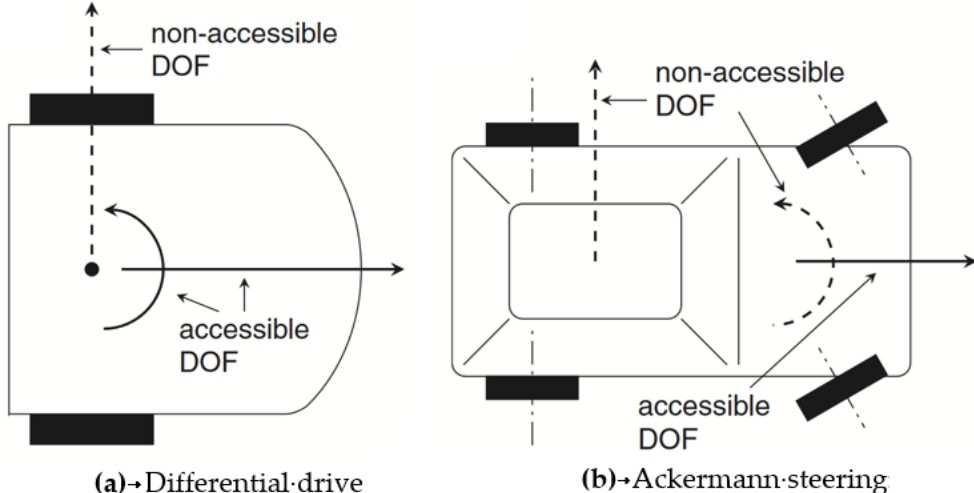

**(a)** Differential drive    **(b)** Ackermann steering

**Figure 3.** Accessible and non-accessible DOF for differential drive and Ackermann steering robots [13].

When one is constructing a customized wheeled UGV, the design will be focused upon a number of determinative characteristics that include, but are not limited to: the number of wheels, the power source of torque (electric batteries, combustion engines, environmental energy harvesting techniques, etc.), the wheel position on the robot's frame, the orientation of the wheels during normal operation, removability, retractability, the wheels' multi-part design (rim and tire combination), material selection, a factor of safety

for failure, size, weight, the center of gravity, and the wheel shape (sphere, cylinder, abstract shape, etc.).

Wheel placement and orientation on the platform change how the actuation of the wheels affects the overall motion. Four-wheeled designs with differential drives, such as the one seen in Figure 4, use a simple design to move forward and backward. However, dynamic equations can model the movement of a robot's movement and the effect of various positions and orientations of the wheels. These models can give useful insights into the various ways to elicit unconventional locomotion, such as the case of the three-wheeled robot depicted in Figure 4.

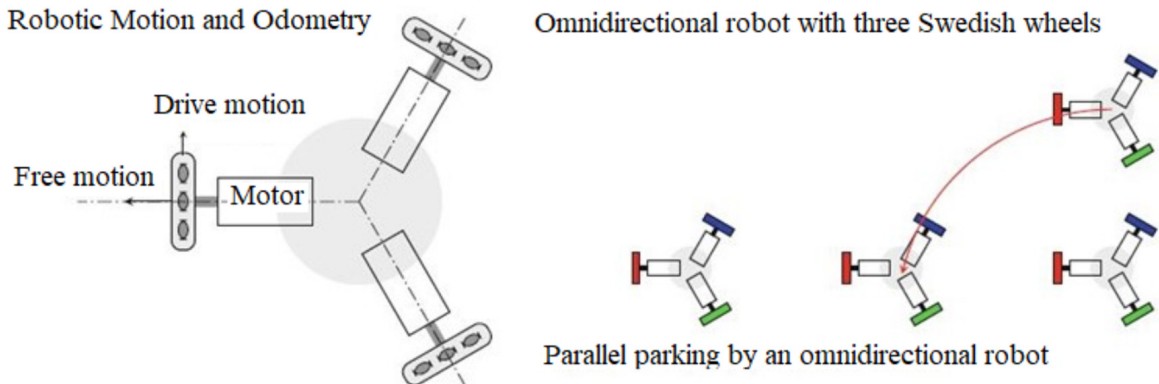

**Figure 4.** A three-wheeled platform that can actuate each motor about a central axis [13].

An important benefit of utilizing or fabricating wheeled platforms stems from the popularity of these systems. The infinite information and resources from previous implementations and in-depth research and experimentation can be significantly insightful [14]. As one of the oldest and most efficient forms of mechanical locomotion, four-wheeled systems are used today for many applications, and thus, have been readily integrated with odometry sensors and techniques and kinematic analysis that can help to improve various aspects of navigation algorithms (see Figure 5a–c). Despite the inherent operational simplicity of the four-wheeled systems, they are unable to efficiently traverse uneven or slippery terrain, which reduces the necessary wheel-to-ground friction.

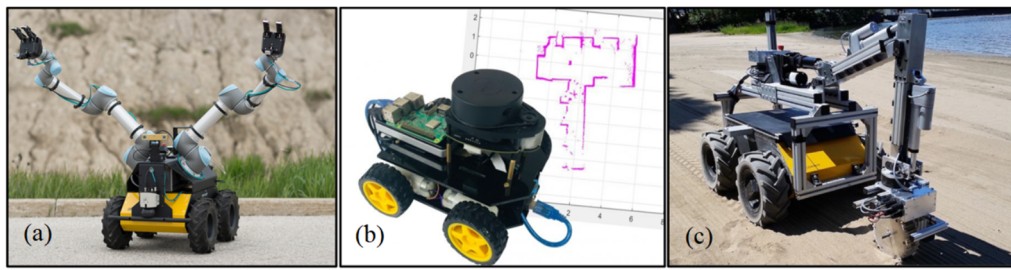

**Figure 5.** Views of (**a**) Husky UGV from Clearpath Robotics equipped with two robotic arms (Clearpath Robotics [15]), (**b**) commercially available wheel-based UGV equipped with LiDAR (SLAM/LIDAR.drone.online), and (**c**) Husky UGV from Clearpath Robotics equipped with custom fabricated soil manipulator crane arm [16].

### 2.1.2. Tracked Vehicles

Tracked UGVs provide a robust alternative to wheeled UGVs [17]. By spreading the weight of the vehicle across a larger surface area, more friction-based adherence to the ground is created. This allows these types of vehicles to operate in uneven and soft-soiled terrain. Although they are slower in speed, the benefits of increased grip and stability are coupled with a stable and simple drive mechanism that allows the tracked vehicles to rotate around themselves (skid-steer locomotion). [18] In most platforms, a track-based

UGV consists of two tracked wheel systems on either side, each with multi-directional control, as seen in Figure 6.

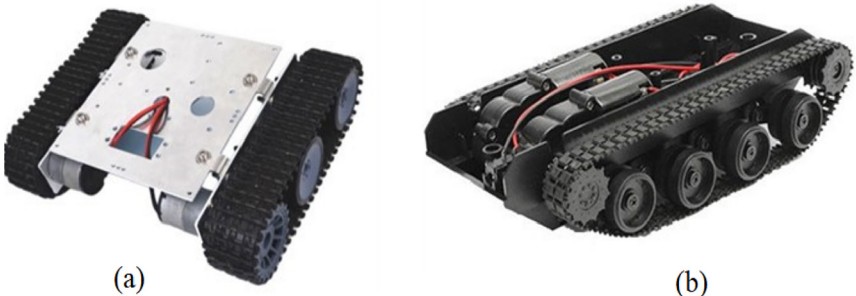

(a)  (b)

**Figure 6.** Track-based UGVs are pre-manufactured and sold by commercial distributors both online and in stores. (Walmart, Online Marketplace, Toys and Electronics, 2022).

Track-based suspension is often installed in tracked vehicles to mitigate the impact of ground-to-track interactions. The installation of spring-mass systems or dampeners allows the tracked vehicles to navigate steep or sudden obstacles without damaging the overall drive mechanisms or the sensor modalities on board (see Figures 7 and 8) [19].

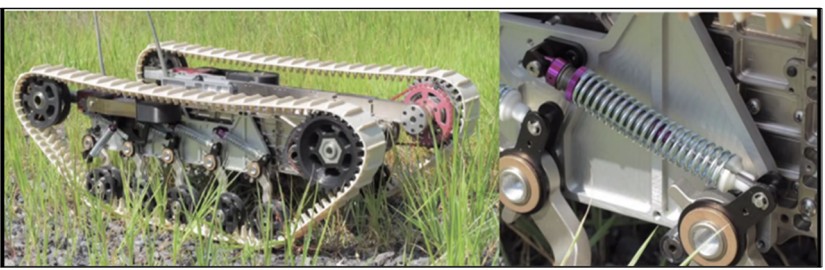

**Figure 7.** The spring-loaded suspension between tank track drives and supports [19].

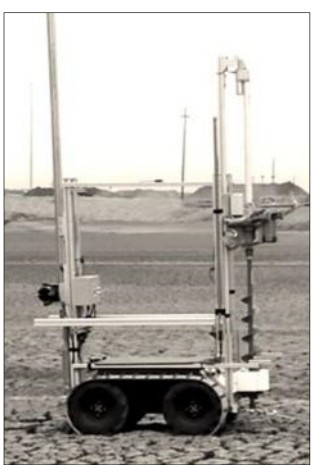

**Figure 8.** Researchers equip a Husky UGV from Clearpath Robotics with tank tracks that fit over the original UGV wheels [16].

Morphing tracks are tank tracks on a UGV that can change their shape or orientation (see Figure 9). The timing of the change in orientation is often up to the designer of the robot, but this capability to change and morph shapes instantaneously gives these drone agents more tools and degrees of freedom to overcome complex obstacles [20].

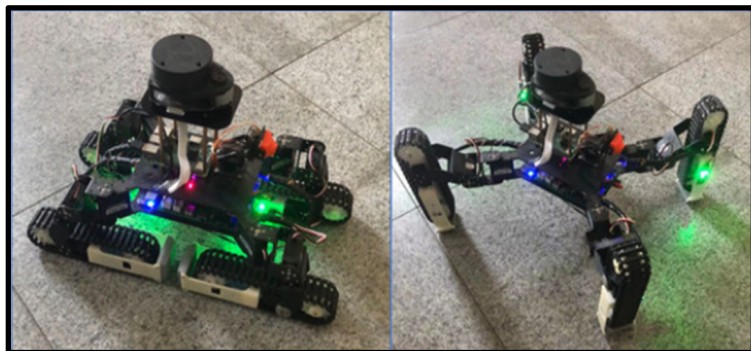

**Figure 9.** View of a morphing-track based robot [20].

### 2.1.3. Bio-Inspired Robots

Bio-inspired robots commonly borrow locomotive mechanisms from the natural world to achieve efficient navigation techniques in environments that are similar to the ones that encouraged the natural evolution of living organisms. Most of the bio-inspired robots are small in size, and hence, they do not have the overload capabilities to support complex autonomous functionalities. However, these robots can be used for auxiliary subtasks of an MAHRS's overall mission. For example, the micro-robots could be used for deploying or reconstructing wireless communication network grids, creating distinctive landmarks or beacons to assist the navigation of the collaborating robotic agents, creating networks for data stations (such as meteorological or hydrological stations), or even assisting in the self-evacuation of trapped humans in subterranean environments. The small size of these robots allows them to carry a single sensor or signaling beacon, while at the same time, it facilitates the storage and deployment of these robots from the larger collaborating robotic agents. Moreover, as nanotechnology, solid state technology, and MEMS advance such robots are becoming increasingly more powerful. The great flexibility of such dynamically deployed networks could be a critical factor for the successful execution of an MAHRS's mission. The bio-inspired robots examined here include legged robots, snakelike, earthworm-inspired, climbing, rolling, jumping, as well as hybrid motion robots.

#### 2.1.3.1. Legged Robots

Bio-inspired robots borrow many mechanisms from nature, but one very common means of locomotion that is used is leg-based propulsion. Leg-based designs can be used for walking, running, or even jumping [21–24]. Leg-based designs can contain any number of legs based on the animals that are simulating, e.g., bipedal ones, quadrupedal ones, hexapods ones, etc. (see Figure 10). Each design achieves a slightly different locomotive goal depending on how many legs are present and how they are arranged.

A spring-loaded scissor jack leg uses multiple joints to accomplish the greater distribution of load across the leg; when they are placed in line and operated by a tank-driven motor, bio-inspired legs can be used even in tank track configurations (see Figure 11) [19].

There are also many approaches to the design of an individual leg. One common realization of a robotic leg includes double-jointed assemblies (one at the hip and one at the knee) [25]. This joint distribution is similar to the leg designs of humans, dogs, cats, etc., ignoring the joint at the ankle. The double-jointed design can be realized with different actuators, both linear and rotational ones, as shown in Figure 12. The benefit of using a linear actuator as the knee joint, known as the pogo stick method, is the simplicity of the system. In this case, there is no geometric work due to the presence of a single angle of the action. The double-rotary joint design, however, is more closely related to the legs of animals. While the design is complicated by the second angle that is introduced, there are more stances, gaits, and types of motion possible with this added dimension of flexibility than there are in the pogo stick design. In Figure 12, views of a robotic leg with one linear

actuator at the knee and a rotary actuator at the hip and a leg with two rotary actuators, one at the hip and one at the knee, are shown [25].

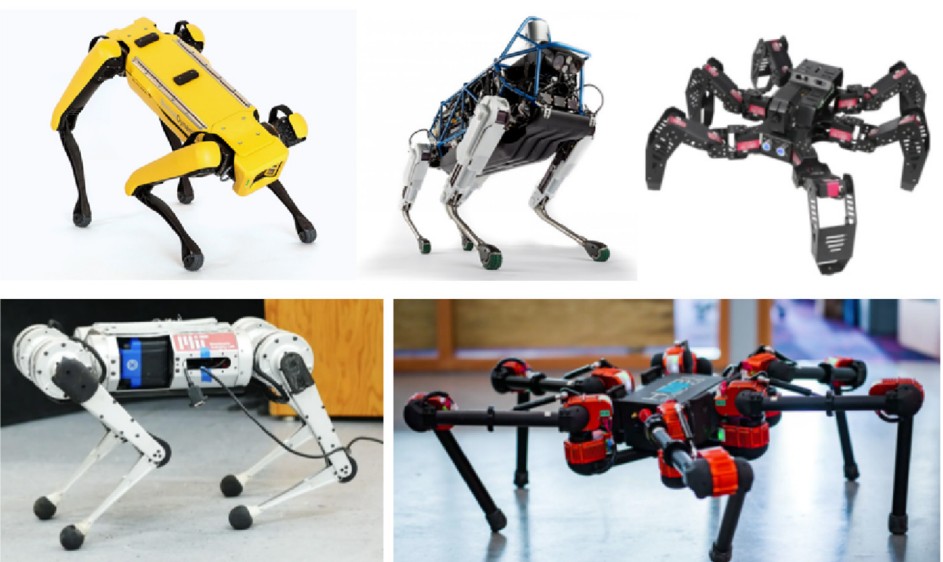

**Figure 10.** View of leg-based robots.

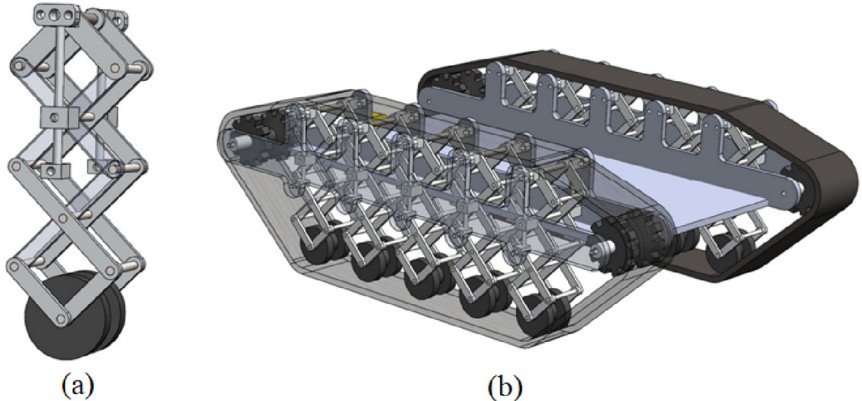

**Figure 11.** Views of (**a**) scissor jack leg assembly and (**b**) scissor jack tank suspension assembly [19].

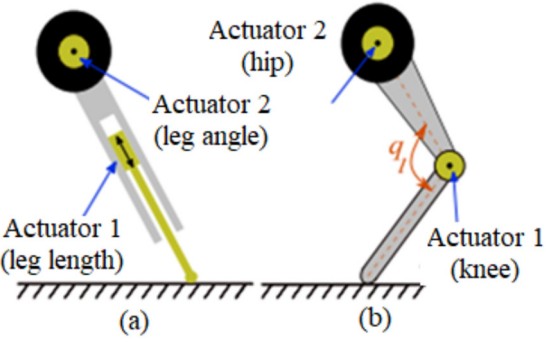

**Figure 12.** Views of (**a**) a robotic leg with one linear actuator at the knee and a rotary actuator at the hip and (**b**) a leg with two rotary actuators: one at the hip and one at the knee [25].

2.1.3.2. Snake-Like Robots

Among snakes, four primary locomotion modes, also called gaits, have been documented: lateral (serpentine) undulation, concertina, sidewinding, and rectilinear motion [26] (see Figure 13). Concertina locomotion involves a longitudinal wave throughout

the snake's body, which winds up in a sort of ribbon shape, and then extends forward, and the snake repeats the process. Serpentine locomotion is the gait that is most associated with snake movement. During this type of locomotion, the body generates a transverse wave from head to tail and uses this periodic oscillation to propel itself forward. Sidewinding locomotion is the fastest way for a snake to travel, as it involves the sideways stretching and contraction of the body, where only a couple of points on the snake's body contact the ground at a time. This type of locomotion is often likened to a sort of 'running' motion in snakes that are capable of it because of their speed and a small amount of relative ground contact. The last type of snake locomotion, rectilinear or caterpillar locomotion, is a linear mode of transport. This locomotion is best used in cramped environments where sideways undulation is not possible due to the restricted space. Longitudinal stretching and compression make this slow locomotion method possible in the same way that a caterpillar moves, with subtle undulating movements [26].

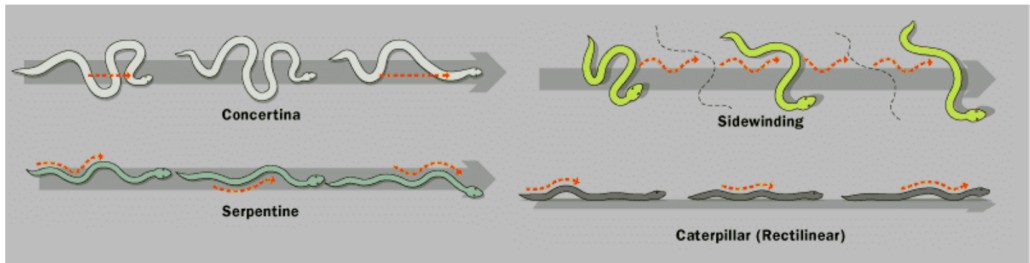

**Figure 13.** Views of four primary modes of snake locomotion: concertina, serpentine, sidewinding, and rectilinear [26].

Different gaits are desired for different engineering goals when one is designing robots with snake-like locomotion properties. Sidewinder locomotion is likely desired for applications where the robot is expected to speedily transverse a loose, sandy environment or an environment where the ground surface is dangerously hot or otherwise corrosive due to short contact duration. Rectilinear locomotion is necessary for locomotion in cramped environments such as inside slender tunnels or pipes [27]. The types of applications for this type of locomotion include search and rescue missions in tight spaces or pipe inspection for non-destructive testing purposes. Serpentine motion has the advantage of being relatively easy to model using simple actuator units. The locomotion takes advantage of traveling transverse waves, and thus, it can be achieved by creating mechanical oscillations across the body of the robot. This type of bio-inspired serpentine motion is used by the robot shown in Figure 14 [27].

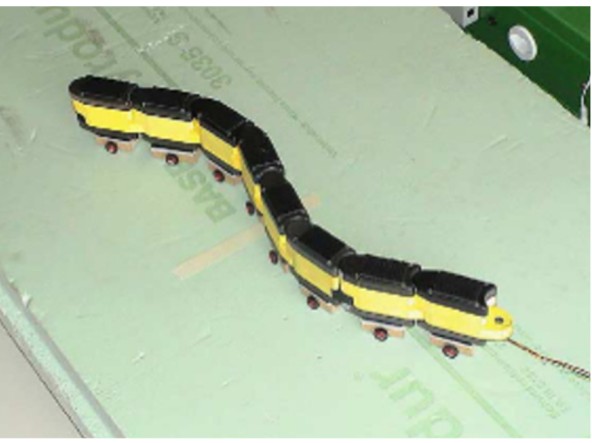

**Figure 14.** Snake-inspired robot with passive wheels controlled by non-linear oscillators which mimic serpentine locomotion [27].

In 2013, Raytheon Sarcos developed a new generation tandem track snake-like robot (see Figure 15), which is suitable for operations in underground mines [28].

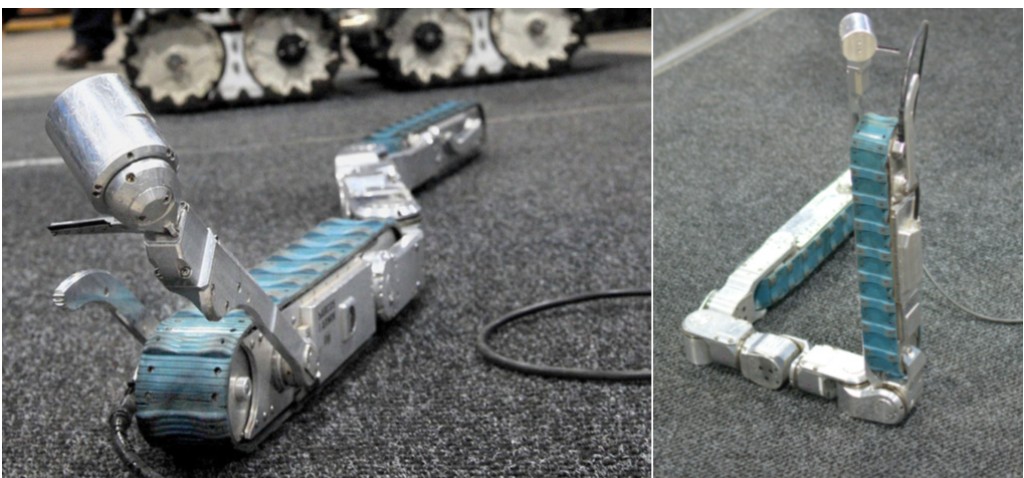

**Figure 15.** Snake-inspired robot developed by Raytheon Sarco for operations in underground mines [28].

### 2.1.3.3. Earthworm-Inspired Robots

Earthworms utilize longitudinal undulation in order to propel themselves forward across surfaces and underground. Their narrow bodies are well adapted to this type of motion, and building robots whose movements are inspired by earthworms has promising applications such as search and rescues, industrial inspections, and medical endoscopy [29].

In order to replicate this method of locomotion, small, linear actuators must be used to replicate longitudinal undulation. As evident in Figure 16, shape memory alloy (SMA) wires are coiled into spring shapes and linked together, then wrapped in a nylon mold. When an electric current is applied to the brass linkages in this SMA skeleton, the spring is heated to induce the Joule effect. This causes the SMA spring to contract, and when the current is shut off, the spring is able to expand gradually, and this process is repeated to create periodical, longitudinal undulating motion [29].

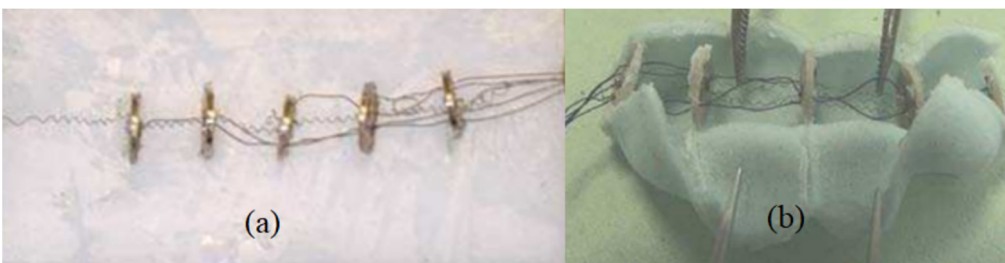

**Figure 16.** View of (**a**) shape-memory alloy skeleton of robotic earthworm design and (**b**) wrapped in a Nylon mold [29].

This SMA skeleton, which is wrapped in the nylon casing, creates the robotic earthworm capable of undulatory travel with four independently controllable modules. Undulatory patterns with a typical frequency of 0.5 Hz is used to create a longitudinal wave that travels across the length of the robot and allows it to propagate through a medium or across a surface. The use of tiny micro-legs on the base of the design allows it to generate enough traction to climb up inclined surfaces with small undulations (see Figure 17) [29].

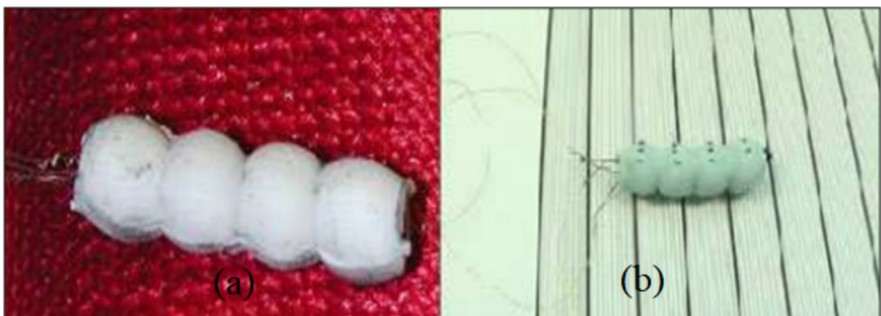

**Figure 17.** View of robotic earthworm design using shape memory alloy locomotion technology (**a**) with and (**b**) without micro-legs [29].

### 2.1.3.4. Climbing Robots

Many natural climbing locomotion instances can be found and studied when one is intending to design a robotic climber. Many animals, such as insects, lizards, and amphibians, can easily climb sheer surfaces. Much of their climbing success can be attributed to the natural adhesive effects of their appendages, their lightweight, and the efficiency of their motion. Geckos, for instance, utilize directionally preferential microfibers on their foot pads to generate a frictional force on vertical surfaces, which counteracts its own gravitational force [30]. This is made possible by preloading each foot placement and dragging downwards to generate adhesion. In order to detach each foot pad, gradual peeling is performed to nullify the adhesion in a similar fashion to that of hook and loop fashioners such as VELCRO®. This process is depicted in Figure 18 [30].

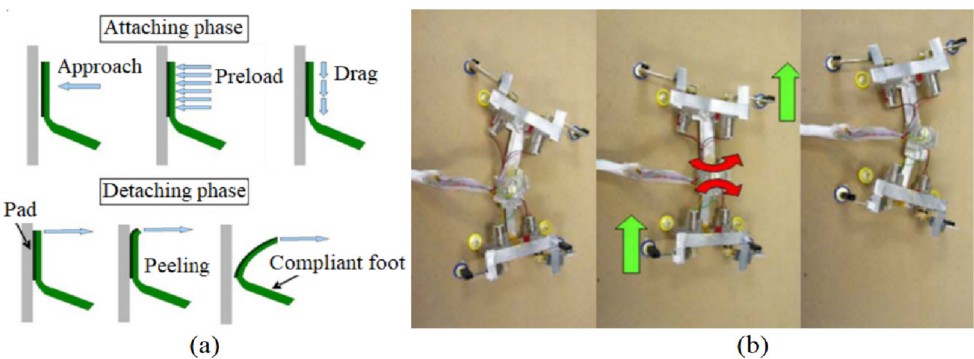

**Figure 18.** Views of (**a**) ideal attaching and detaching phase diagram of gecko foot used while climbing and (**b**) gecko-inspired climbing robot with flexible foot and hip joints and Silly Putty® adhesive foot pads [30].

When smaller, faster robots are required, bio-inspiration may be drawn from the insect or arachnid classes of the animal kingdom. In order to maximize the effectiveness of the climbing robot, considerations need to be made as to what makes their natural counterparts such effective climbers. A high contact-to-weight ratio is desired for adhesion, which is achieved by a low mass and a high leg count. This allows insects such as cockroaches to be very efficient and very fast climbers. Micro-actuators and shape memory alloys can be used to replicate insect-based climbing in robotic applications, such as the one shown in Figure 19 [31].

A four-legged climbing robot "Limbed Excursion Mechanical Utility Robot" called LEMUR was designed and prototyped by NASA's Jet Propulsion Laboratory and several other institutions. LEMUR uses grippers that are armed with hundreds of tiny, sharp hooks that enable it to climb rock walls and navigate underground spaces [32]. In Figure 20, a view of this climbing robot is shown.

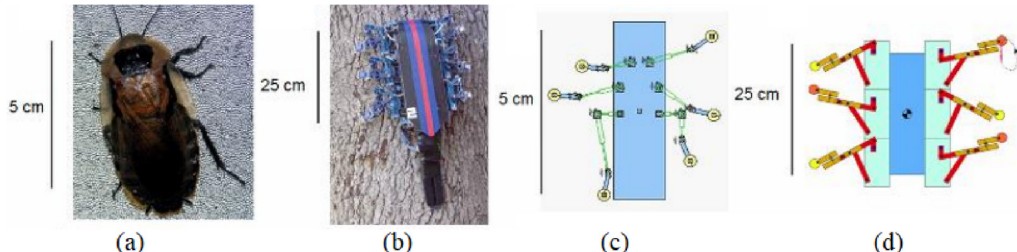

**Figure 19.** Views of (**a**) climbing insect, (**b**) insect-inspired robot, (**c**) numerical insect model, and (**d**) numerical robot model. The numerical models were used to compare normal reaction forces and investigate leg configuration, trajectory, and compliances [31].

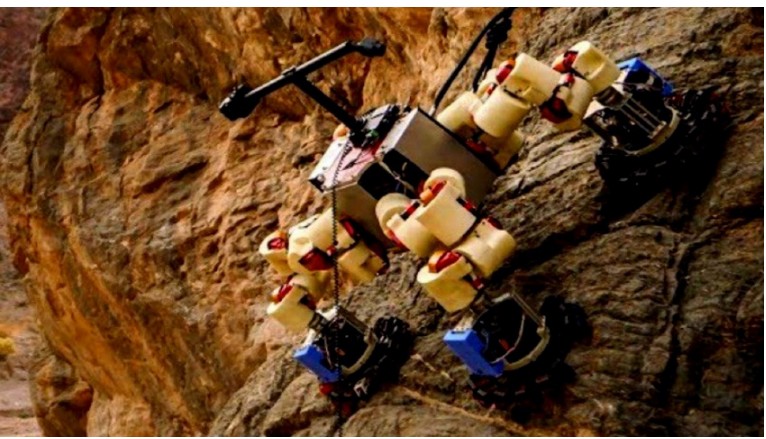

**Figure 20.** View of a four-legged climbing robot [33].

2.1.3.5. Rolling Robots

Some robots are designed to be able to move via rolling, which can be especially effective and energy saving when they are traversing certain terrains where gravity aids the direction of travel. Some of them are designed to be completely globular, relying on rolling as the sole method of maneuvering. Rolling robots that use rolling as their sole locomotion method can be classified into three control categories: control by the center of mass, control by variable gyrostatic momentum, and control by deformation [34]. Other hybrid designs are designed to be able to walk and roll, which makes them more similar to the animals they draw inspiration from, such as isopods, armadillos, and rolling spiders (see Figure 21). These robots often involve a process that contracts them into a spherical shape, which is a process called conglobation.

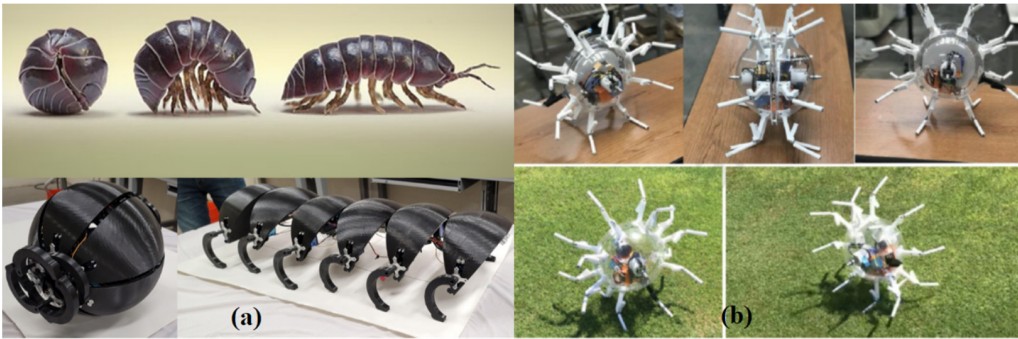

**Figure 21.** View of (**a**) rolling isopod-based robot demonstrating the conglobation process [34] and (**b**) golden-wheel-inspired rolling robot [24].

### 2.1.3.6. Jumping Robots

Some bio-inspired robots attempt to emulate the jumping motion of many animal species, such as frogs and kangaroos. This method of locomotion may be beneficial to the robot when it is navigating terrain with obstacles that are impossible to traverse with a traditional rolling wheel locomotion method. One of the difficulties of the jumping method of locomotion is designing a robot that remains perfectly balanced throughout each consecutive jump, and extensive testing is needed to find the robot's center of gravity. Many jumping robots, such as the kangaroo bot shown in Figure 22, utilize movable counterbalances to simulate the tail. This keeps the robot from over-rotating or under-rotating during each jump and sets it up to proceed with the subsequent jump [35].

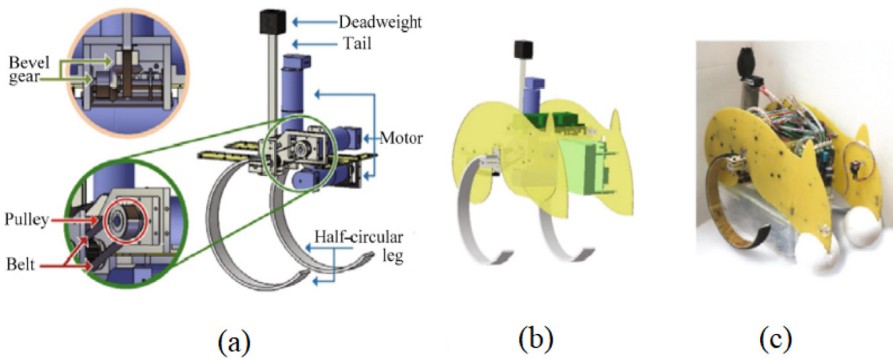

**Figure 22.** View of a jumping-based bio-inspired kangaroo robot; (**a**) transmission system of kangaroo robot, (**b**) CAD drawing of kangaroo robot, and (**c**) photo of kangaroo robot [35].

### 2.1.3.7. Hybrid Motion Bio-Inspired Robots

While many animals have achieved the mastery of certain types of motion, there are also many animals that are capable of multiple modes of transportation. Whether it is climbing and walking, walking and swimming, climbing and flying, or many other combinations, there are natural counterparts to observe for any desired type of locomotion one might have for robotic design. One of the most desirable qualities of a locomotive robot is the ability to traverse both land and water. These amphibious robots can be modeled with inspiration from many amphibious animals, such as otters, crocodilians, salamanders, marine iguanas, and many more. Figure 23 shows a prototype for one such amphibious robot design. Instead of utilizing swimming and walking locomotion modes such as salamanders or crocodilians do, this prototype design uses actuated fins and a wheel that doubles as a propeller when it is underwater [36].

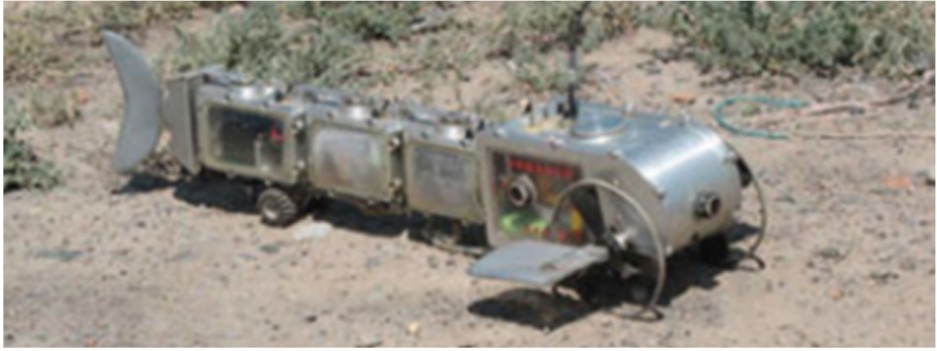

**Figure 23.** Mechanism design of AmphiRobot-II; prototype of the amphibious robot in the field test [36].

## 2.2. Drones

Recently, the mining industry has shown an increased interest in using drones for routine operations in surface and underground mines [4–6]. Shahmoradi et al. provided a comprehensive review of the current state of drone technology and its applications in the mining industry. Their study presented the configurations, specifications, and applications of commercially available drones for mining applications [4]. In the following subsections, different drone configurations are discussed including single- and multi-rotors, thrust-vectoring aerial vehicles, moving-mass controlled vehicles, coandă effect drones, flapping, winged drones, airships/blimps, and cyclocopters.

### 2.2.1. Single Rotors/Helicopters

Single-rotor drones are useful in the applications of surveying and construction. Due to the greater battery efficiency of powering a single propeller, mono-rotors can lift heavier payloads than multi-rotor drones of the same weight can (see Figure 24). This also creates faster drones, although at the expense of flight stability and directional control fidelity. The biggest challenge to designing a drone with a single rotor is controlling the horizontal flight direction. Using a single rotor enables very easy and efficient vertical flight, but translating itself sideways in a specific direction is difficult to achieve when only a single rotor is used [1,37].

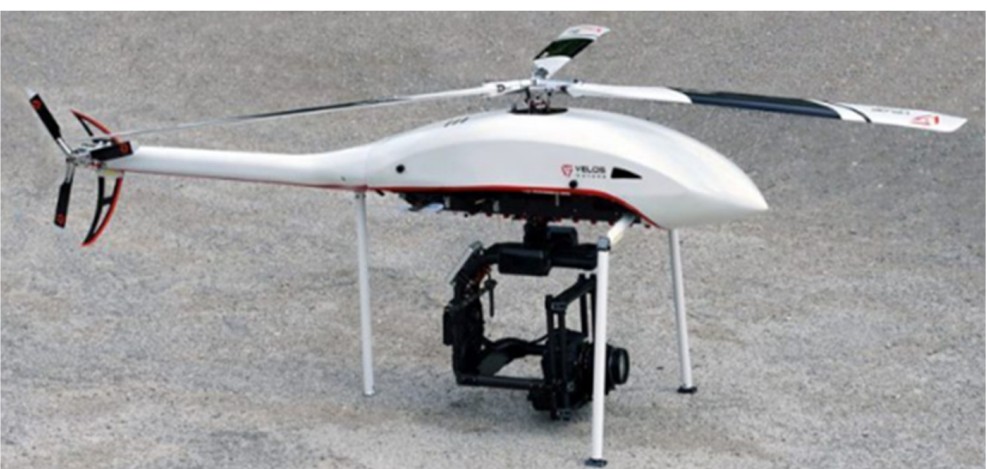

**Figure 24.** Single-rotor helicopter-style drone used for surveillance [37].

### 2.2.2. Ducted Fan/Thrust Vectoring

The challenge of achieving directional horizontal maneuverability can be addressed by utilizing thrust vectoring. Thrust vectoring is a process by which an aerial vehicle can manipulate the direction of thrust from its motor. A mono-rotor drone can accomplish this by including actuated fins below the propeller, which alter and divert the airflow in the desired direction. The mono-rotor drone shown in Figure 25 utilizes this technique and a ducted fan to enable high-speed, controllable, multi-directional flight using a single propeller. The ducted fan enables a more efficient, higher velocity thrust from the propeller by housing it in a cylindrical shroud [1,3,38].

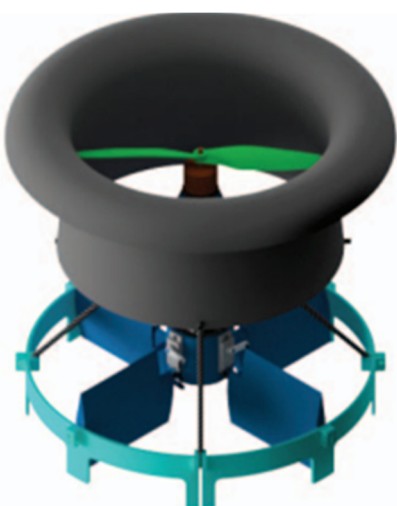

**Figure 25.** Design of single rotor drone, complete with ducted propeller, four actuated fins, and a structural/landing support [38].

### 2.2.3. Moving-Mass Controlled (MMC) Drones

Moving-mass controlled (MMC) drones utilize a unique method of directional control that involves changing the drone's center of gravity (see Figure 26). Unlike other single-rotor drones, which often use swiveling tilt rotors to control the horizontal movement of the drone, MMC drones often have a fixed rotational axis for the propeller and instead use linear shifts in weight for directional control. Linear actuators, often moving in two spatial dimensions, provide the required shift in the center of mass to achieve the desired directional flight change [1,39].

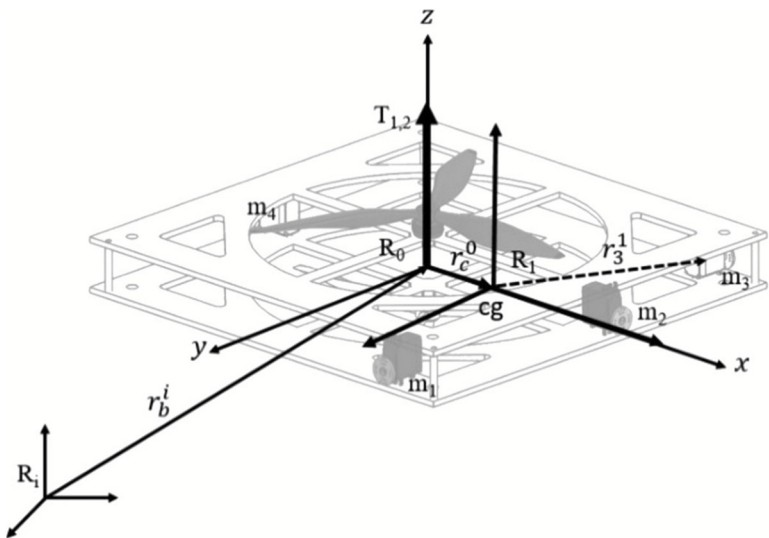

**Figure 26.** Single-rotor drone controlled by two-dimensional moving masses which direct the drone's motion by controlling the center of gravity [39].

### 2.2.4. Coandă Effect Drones

The Coandă effect, named after the Romanian inventor Henri Coandă, is the tendency of a fluid jet, such as air, to stay attached to a convex surface [1]. This effect can be seen when one is observing the fluid flow around a curved airfoil, causing the flow direction to be altered by the convex surface geometry [40,41]. An illustration of this effect can be seen in Figure 27.

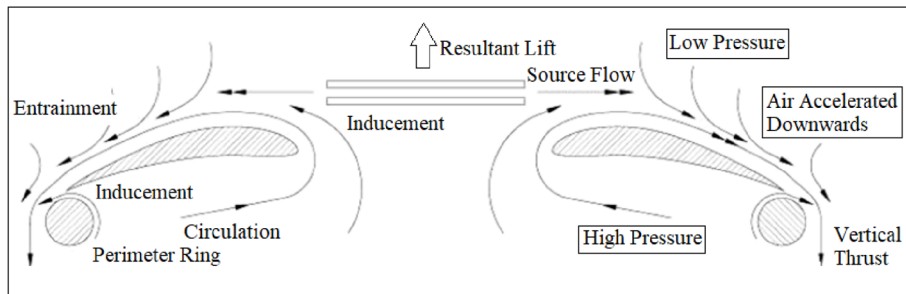

**Figure 27.** Illustration of the Coandă effect to produce resultant lift as air passes through the center of an airfoil ring and increases velocity as it travels around [41].

This effect is used for certain designs that alter the airflow direction and speed for cooling, such as the Dyson Air MultiplierTM fan, or lift, such as the Aesir Coandă drone shown in Figure 28 [41].

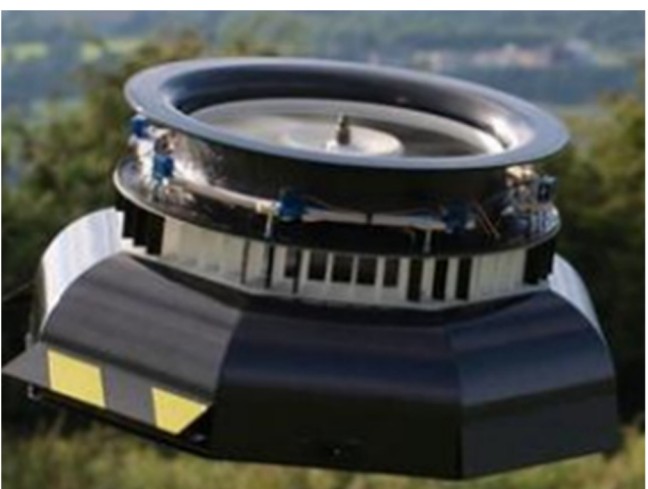

**Figure 28.** Aesir drone utilizing a single rotor and the Coandă effect for flight [41].

### 2.2.5. Multi-Rotor Drones

Besides single-rotor drones or drones that move without rotors, several designs of drones adopt two or more rotors that enable better directional control. In the following subsections, a brief overview of bi-rotors, tricopters, quadcopters, and pentacopters is presented.

### 2.2.5.1. Bi-Rotors

As opposed to the more conventional quadcopter design, it is possible to design a bi-rotor drone with tilting rotors to enable directional control. The primary reason that quadcopters are used more often than bi-rotor drones is the appeal of utilizing fixed rotors while still being able to maneuver the drone forwards and backwards, left and right, as well as making it rotate. The bi-rotor does not have this flexibility with fixed rotors, and balance and symmetry must be sacrificed when one is opting the two-rotor configuration. It is worth noting that this configuration still provides more stability with fewer workarounds than the mono-rotor configuration does [1].

One of the main advantages of using a bi-rotor drone design over a quadcopter one is battery efficiency. The tilting mechanisms on each rotor enable the drone to be controlled without much loss of control or stability, but powering only two rotors facilitates significantly longer flight times, nearly doubling the maximum flight duration from about a half hour (as typical in quadcopter flights) to fifty minutes in the case of the V-Coptr Falcon by Zero Zero Robotics, as shown in Figure 29 [42].

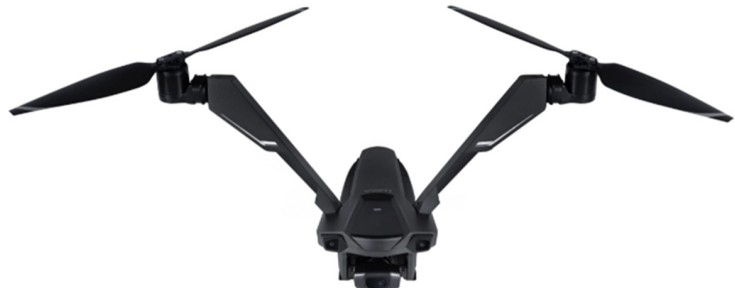

**Figure 29.** V-Coptr Falcon by Zero Zero Robotics with two tilting rotors [42].

2.2.5.2. Tricopters

Tricopters, such as the one shown in Figure 30, are drones with three rotors as opposed to four or more. These drones are vastly less popular than quadcopters are for a few key reasons. While reducing the number of rotors from four to three may reduce the weight of the drone, the loss of thrust imposes limitations. The tricopter has a longer battery life than quadcopters do due to the need to power one less rotor, but not by a large enough margin to make up for the many downsides to three-rotor control. The main reason that tricopters have a hard time competing with quadcopters is the difficulty of control arising from the frame's asymmetry. While quadcopters have four-way symmetry and intuitive solutions to controlling pitch, yaw, and roll, the tricopter does not. Due to the asymmetry of the tricopter, flight control is much more complex than it is with the quadcopter. As a result, numerous studies have been conducted to devise mechanical solutions to controlling altitude, pitch, yaw, and roll. Altitude control and pitch control are relatively straightforward with the tricopter, but unusual combinations of rotor power and servo motors are required to control the drone's roll and yaw. As an example of these complexities, some controlling techniques are illustrated in Figure 31 [1,43].

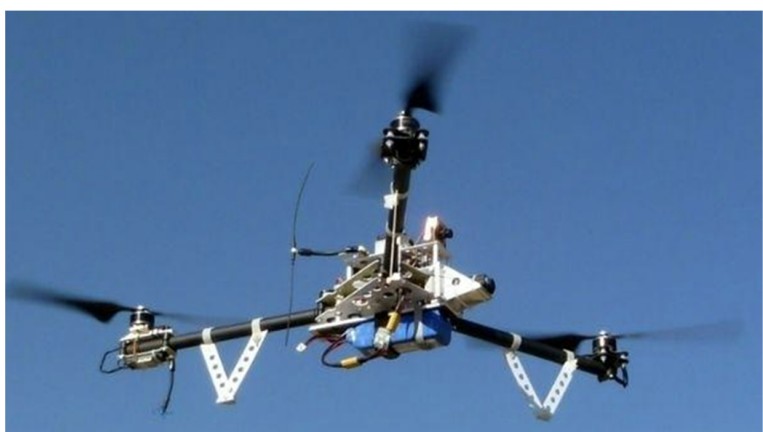

**Figure 30.** Tri-rotor drone in flight [43].

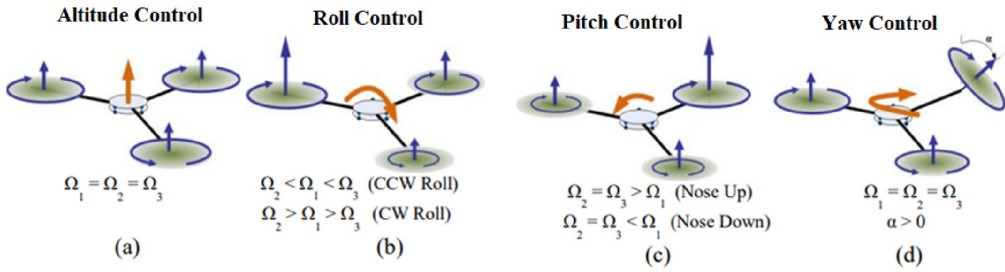

**Figure 31.** Tri-rotor drone control strategies for (**a**) altitude; (**b**) roll; (**c**) pitch; (**d**) yaw [44].

### 2.2.5.3. Quadcopter

Quadcopters with four individually controlled rotors are the most widely used design for drones. The popularity of this design comes from the ease of control granted by the two axes of symmetry, allowing for pitch, yaw, and roll to be controlled with ease. While the battery efficiency is less appealing compared to the drones with less than four rotors, most commercial quadcopters are often able to sustain flights in the range of thirty minutes. Figure 32 shows a custom-made quadcopter based on a commercial racing frame kit. Common applications for commercial quadcopters include security and surveillance, topographical mapping, emergency responses, infrastructure inspections, and even drone racing [45].

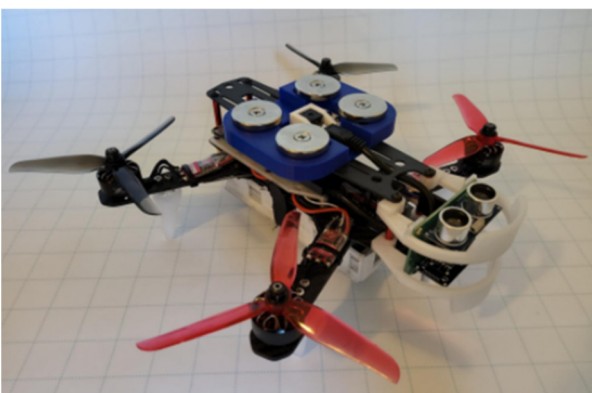

**Figure 32.** Semi-autonomous precision-docking quadcopter constructed from a racing drone frame with custom sensors and flight controllers [45].

### 2.2.5.4. Pentacopters and Others

While quadcopters are considered to be the standard drone design for most applications, drones with a higher rotor count exhibit advantages for specific applications despite the disadvantages of an increased weight, size, and cost of the drone, as well as the decrease in battery life and flight time. Drones with a large number of rotors come with the benefit of a higher weight-lifting potential. Although the weight of multi-rotor drones is larger, the added rotors enable it to have a higher overload capacity. As a result, hexacopter and octocopter drones are often used in applications that require them to lift larger payloads, including delivery drone applications such as the hexacopter shown in Figure 33 [1,46].

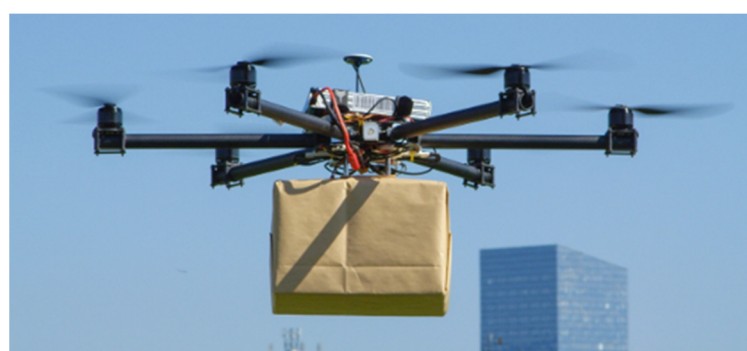

**Figure 33.** Hexacopter drone used for package delivery [46].

While four-, six-, and eight-rotor drones have many commercial applications, five-rotor pentacopters are very rarely used. This is due to the greatly increased difficulty of piloting a drone with an odd number of symmetries, and the development of piloting algorithms for such symmetries lag compared to those of other drones. Some pentacopters utilize quadrilateral symmetry, with the fifth rotor placed in the design center (see Figure 34a)

or used as a frontal propeller (Figure 34b), which allows them to be piloted in a very similar way to traditional quadcopters [1].

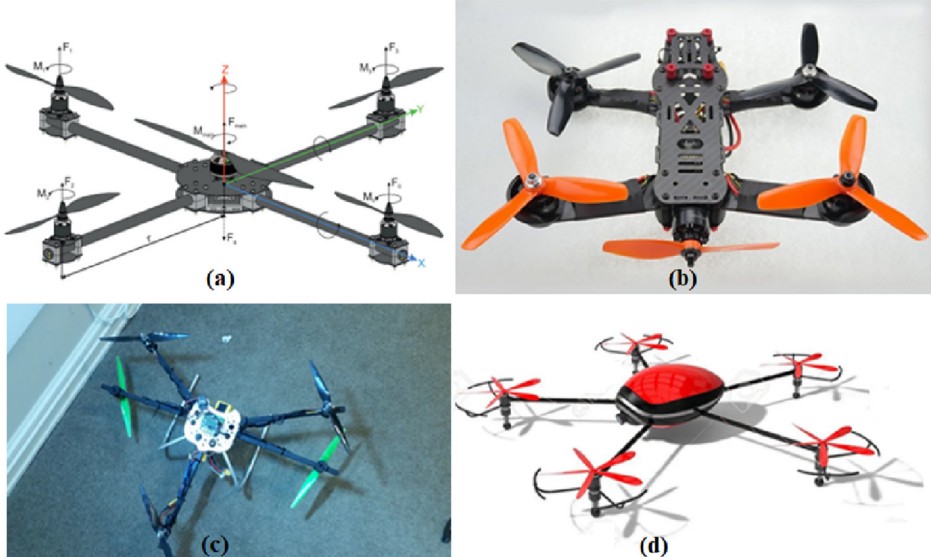

**Figure 34.** Various pentacopter designs: (**a**) central upwards rotor on experimental drone design [47], (**b**) front-facing propeller on Foxtech Screamer Racing Pentacopter [48], (**c**) experimental 'dragonfly' design with four front rotors and a tail for pitch control [49], and (**d**) a purely hypothetical equidistant rotor design [50].

Although pentacopters are often deemed to be impractical for flight due to the difficulty of controlling them, six-rotor and eight-rotor drones are commercially available for purposes such as surveying and payload delivery (see Figure 35). One of the benefits of six- and eight-rotor drones, besides their higher lift potential, is rotor redundancy. This means that in the event of a rotor failure during flight, the drone can remain airborne, as opposed to a quadcopter, which becomes inoperable in the event of a single rotor failure [1].

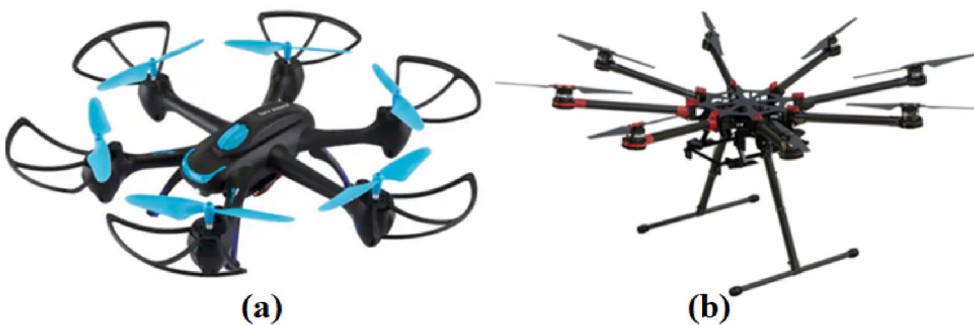

**Figure 35.** Multi-rotor drones include commercial versions of (**a**) a hexacopter by Sky Rider [51] and (**b**) an octocopter by DJI [52].

### 2.2.6. Flapping Wings

Flapping wing drones are inspired by birds or insects and imitate them with the help of mechanical wings to gain altitude and fly in the desired manner (see Figure 36). Flapping wing drones have the potential to offer better maneuverability advantages over those of other drones of the same size [53–63]. Small insect- and hummingbird-inspired drones could potentially be capable of Vertical Takeoff and Landing (VTOL). A disadvantage of flapping wing drones is the complexity of their actuation mechanism.

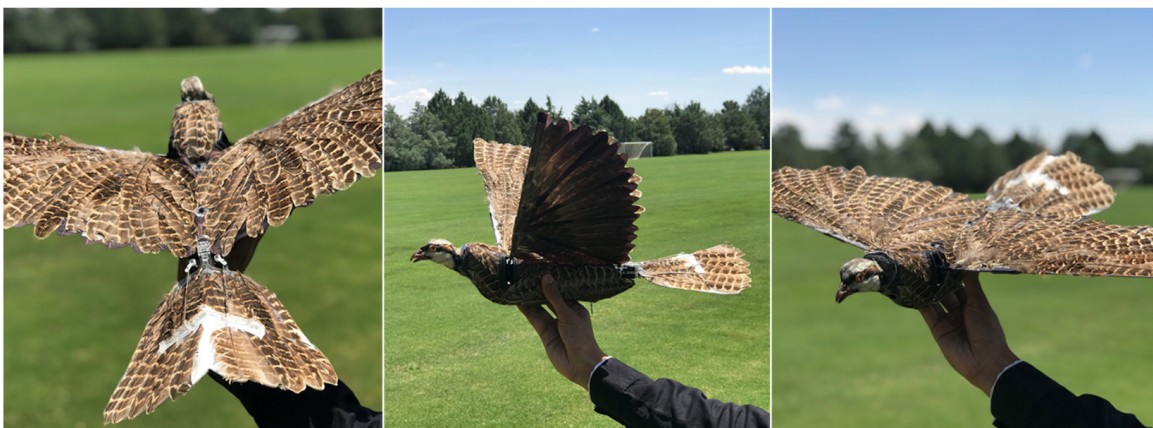

**Figure 36.** A flapping wing drone that took bio-inspiration from dragonflies [59].

### 2.2.7. Airships and Blimps

Unmanned airships rely on lighter-than-air buoyancy for lift, as opposed to rotors or flapping wings. These drones are required to be much larger than traditional quadcopters or flapping wing drones are due to the requirement of large volumes of lifting gas (see Figure 37). A major advantage of airships is that they require a very small amount of battery power to stay gas aloft, and hence, they endure for longer. On the other hand, airships have the major drawback of a low payload capacity despite their size. The large size of airships also creates a lot of drag, which hinders their performance in air currents and prevents them from achieving high speeds [1,64].

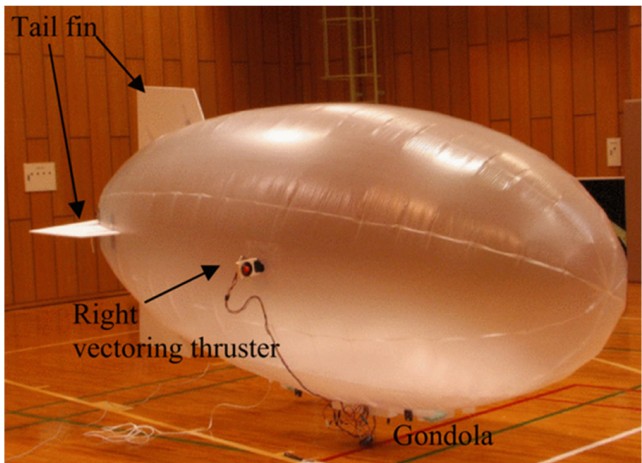

**Figure 37.** An autonomous airship that uses thrusters and tail fins for control [65].

### 2.2.8. Cyclocopters

Cyclocopters generate lift through an arrangement of airfoils that rotate continuously to control lift and thrust (see Figure 38). Configuring the pitch and rotation of the airfoils enable the control of the cyclocopter's movement [66]. Cyclocopters can come in several configurations, including twin and quad configurations, and they are also capable of VTOL [1,3]. However, two major drawbacks of cyclocopters are their mechanical complexity and the complexity of their control.

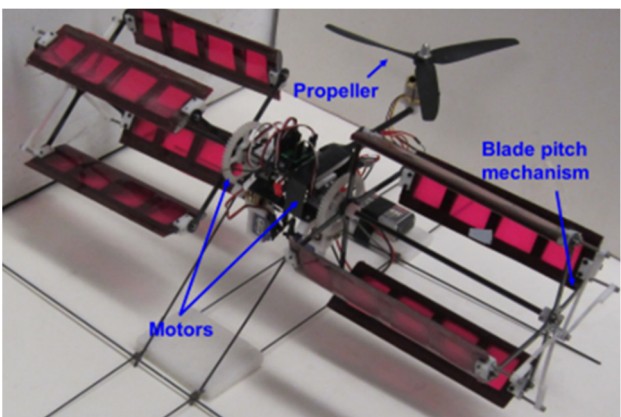

**Figure 38.** A twin cyclocopter that uses the rotating blade pitch mechanism to produce lift [66].

*2.3. Hybrid Drone–UGV Systems*

Hybrid single-agent systems where a UGV can convert to or simultaneously function as a drone, and vice versa, are a popular alternative to expand the capabilities of a single robot. Common examples include tank quadcopters, walking/jumping drones, rolling–flying drones, climbing–flying drones, flying UGVs, and transformable drones.

### 2.3.1. Tank Quadcopters

One type of unmanned hybrid aerial–terrestrial vehicle is a tank quadcopter, i.e., a quadcopter fitted with powered tank treads, allowing for both all-terrain ground travel and aerial travel. This option allows the unmanned vehicle to navigate and execute missions in a large range of environments. The treads would allow the drone to easily traverse difficult terrain and could be designed to be relatively lightweight compared to a drone with multiple wheels. Treads that wrap all the way around the drone, such as in Figure 39, would also allow the drone to continue ground travel upside down if the vehicle happens to be turned over.

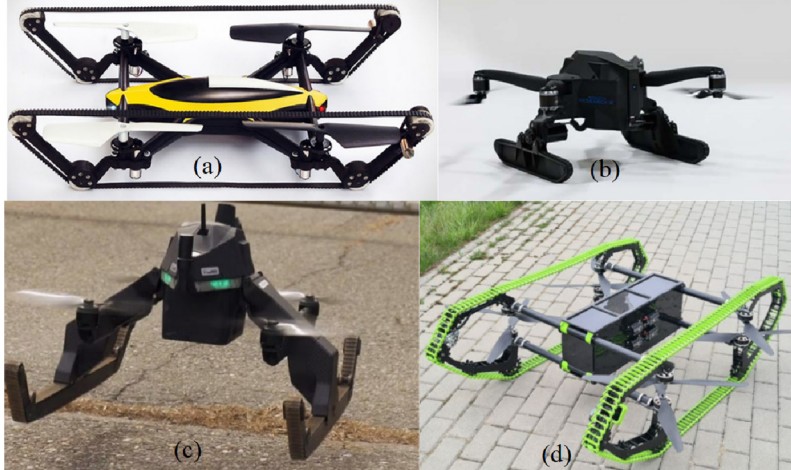

**Figure 39.** Views of tank quadcopter complete with powered treads for terrestrial travel and four rotors for aerial travel, (**a**) [67], (**b**) [68], (**c**) [69], and (**d**) [70].

### 2.3.2. Walking/Jumping Drones

Walking and jumping drones are both potential concepts for hybrid aerial–ground-based robots (see Figure 40). They would both allow the drone an alternative means of locomoting on the ground; this allows the drones to traverse terrain without flying, which can save the battery power. A drawback to walking/jumping drones is that the drone has

to use additional battery power to lift the added weight of the walking/jumping mechanism [71].

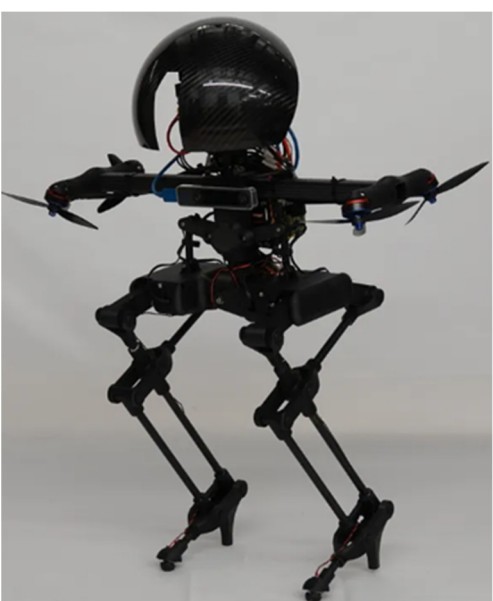

**Figure 40.** LEONARDO, the walking, hopping, flying drone, is modeled after birds that can both fly and hop in order to navigate telephone lines [71].

### 2.3.3. Rolling–Flying Drones

Another interesting concept for a hybrid aerial–ground-based robot is a rolling–flying drone such as the one in Figure 41. It is similar in concept to a flying UGV; however, this concept focuses more on the drone aspect than it does on the UGV aspect. A rolling drone is advantageous because it has the ability to roll on terrain, whenever possible, which allows it to use less battery power while it is travelling. A drawback of rolling–flying drones is that flying requires more power because it has to lift the additional weight of the rolling mechanism.

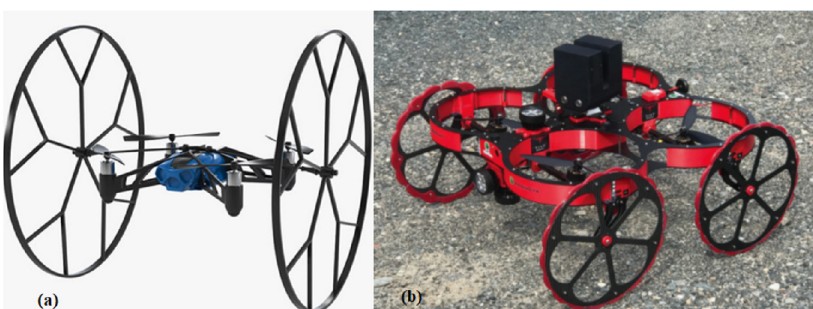

**Figure 41.** (**a**) Parrot MiniDrone Rolling Spider is an example of a quadcopter drone that utilizes both flying and rolling mechanics of motion, fitted with wheels that allow it to roll across the ground and up walls [72] and (**b**) Tilt Ranger drone with V-SCAN3D scanner (Clickmox Solutions) [73,74].

### 2.3.4. Climbing–Flying Drones

Climbing–flying drones are designed with the ability to climb obstacles or walls if they landed on or near them. The climbing functionality allows the drones to climb over difficult terrain or obstacles instead of flying over the obstacles, and hence, it reduces energy consumption. VertiGo, shown in Figure 42, uses a bi-rotor configuration to assist the drone with climbing walls [75].

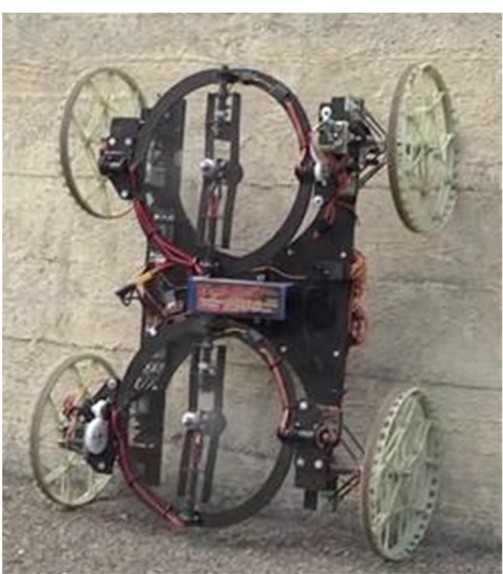

**Figure 42.** VertiGo, a drone developed by Disney's research team, is an example of a drone that is capable of scaling walls by using four gripping wheels and a bi-rotor drone, which allows flight and maintains traction on walls [75].

### 2.3.5. Flying UGVs

Flying UGVs are another possible hybrid aerial–ground system. Unlike rolling–flying drones or tank quadcopters, the UGV is the primary focus of this type of transformable autonomous agent (see Figure 43). The flight capability of the UGV allows the agent to fly over terrain that is impassable to the UGV's locomotion mechanism. However, a major drawback of a flying UGV is the higher power consumption due to the rotors having to lift the entire weight of the UGV. This can be mitigated by keeping the UGV's flights short in duration.

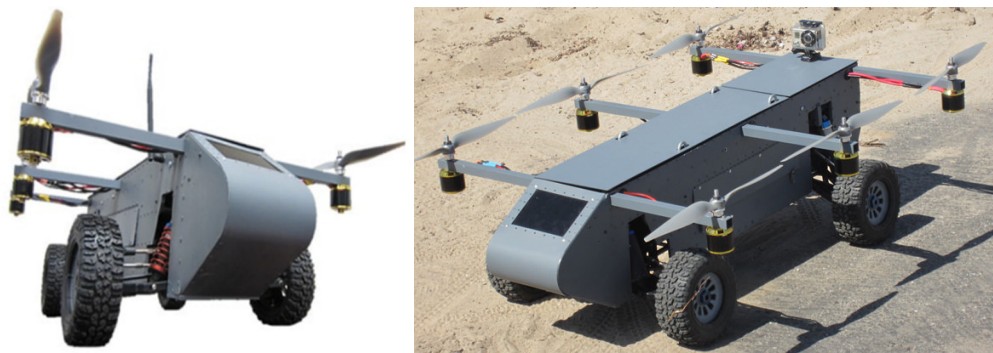

**Figure 43.** Advanced Tactics Panther Drone is one of the first unmanned ground vehicles (UGVs) capable of flight in addition to terrestrial travel. The UGV uses four powered wheels, as well as four rotors that act as a quadcopter with pitch and roll control [76].

### 2.3.6. Transformable Drones

Transformable robots are designed to be able to alter their configuration or shape. Many of them consist of smaller modular components that can be assembled in various manners, as shown in Figure 44. Others have hinges or sections that can move dynamically, allowing the drone to change shape. The ability to transform can be beneficial since it allows the drone to adapt its configuration according to the unique needs of the environment in which the drone operates, allowing it to adapt to dynamic environments.

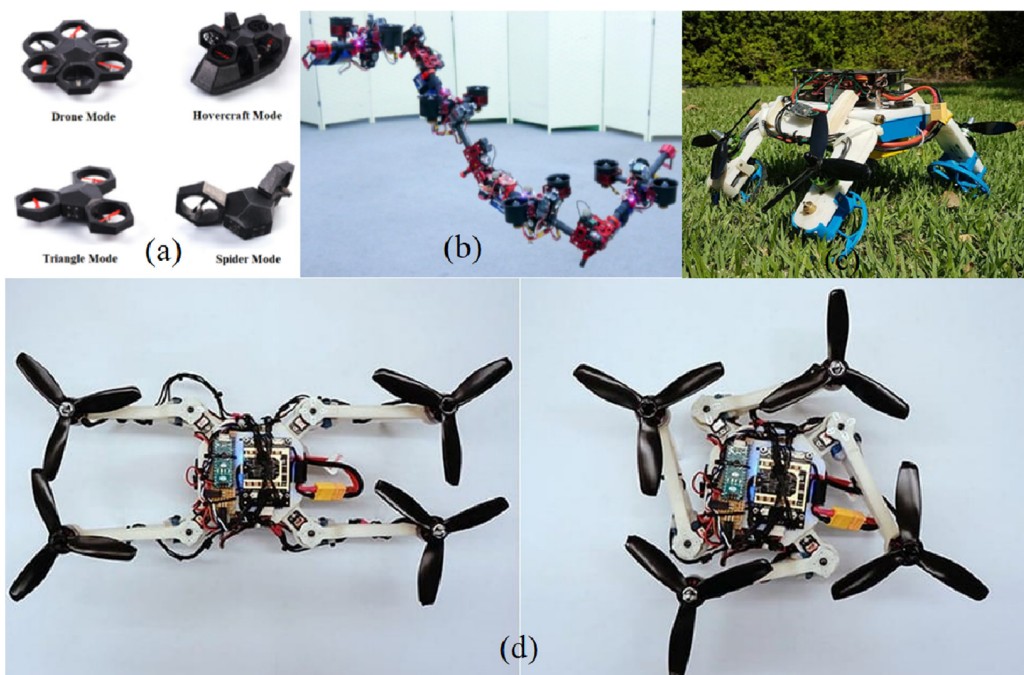

**Figure 44.** View of (**a**) Airblock transformable drone is an example of a modular transformable drone that consists of a central core and six modular rotor blocks, which can be rearranged into different flight and ground travel modes, as seen above [77], (**b**) DRAGON, which is somehow an acronym for the tongue-twisting description "Dual-Rotor Embedded Multilink Robot with the Ability of Multi-Degree-of-Freedom Aerial Transformation," [78], (**c**) this drone can transform into a tiny car to slide under small gaps [79], and (**d**) morphing design for quadrotors [80].

## 3. Cooperative UGV–UAV Systems

There have been recent efforts to increase the degree of automation and frequency of data collection for GPS-denied environment applications using UGV–UAV systems. The combination of drones and UGV, where a drone can be deployed by a UGV, and vice versa, can be applied for search and rescue missions in harsh environments. However, the current practice of the missions in these environments is traditionally performed, which is manual, costly, and time consuming. Developing vision-based mobile robotic systems that are aware of their surroundings and capable of autonomous/semi-autonomous navigation are becoming essential to many search and rescue applications, specifically for underground environments.

Applying a single unmanned vehicle in an underground environment mission incurs a series of performance issues. One major problem is inefficient navigation and agility in indoor and cluttered spaces with many obstacles and barriers, where some places are inaccessible by a UGV. To provide a solution to this problem, a team of mobile robots which integrate drones and UGVs can be applied. In these systems, an unmanned ground vehicle will semi-autonomously/autonomously navigate underground spaces while carrying a drone. The drone will act as an external eye for the UGV, observing the scenes in underground environments that are inaccessible to the UGV.

### 3.1. Drones Deployed by UGVs

One possibility for drone and UGV integration is to have drones that are deployed by UGVs. This option is advantageous because it allows the UGV to carry drones to the mission; this allows the drone to have a longer flight duration in the mission area because the drone does not have to use its battery while it is in transit to the mission area. The UGV can also act as a charging station for deployed drones; UGVs equipped with housing for drones protect the drone from the elements. Drones deployed by UGVs can also aid with the UGV's operations. The drone can be used to assist with mapping obstacles for

the UGVs navigation system. The drone can also be used to directly assist the UGV with navigating obstacles. One system that was created used a drone to tether a UGV's cable to difficult terrain; the UGV was then able to use a winch system to pull itself over difficult terrain [81].

Planck AeroSystems and Stealth Technologies have developed a capability for autonomous UGV that could deploy drones autonomously. The UGV has been developed for perimeter security applications in sectors such as defense, transport, government and utilities, and energy (see Figure 45a) [82]. A Russian combat robot also has been designed for patrolling without human assistance, navigating its way across a 100 km route and working with a swarm of drones (see Figure 45b) [83]. The robotic dog/grasshopper from Boston Dynamics has been modified to carry and deploy a drone (see Figure 45c) [84]. In 2014, a team of researchers from the University of Waterloo developed a control system that enabled a drone to land safely on a UGV (see Figure 45d) [85]. Cocchioni et al. worked on the interaction between a UGV and a drone to extend the endurance of the aerial system thanks to a novel landing/recharging platform (see Figure 45e) [86].

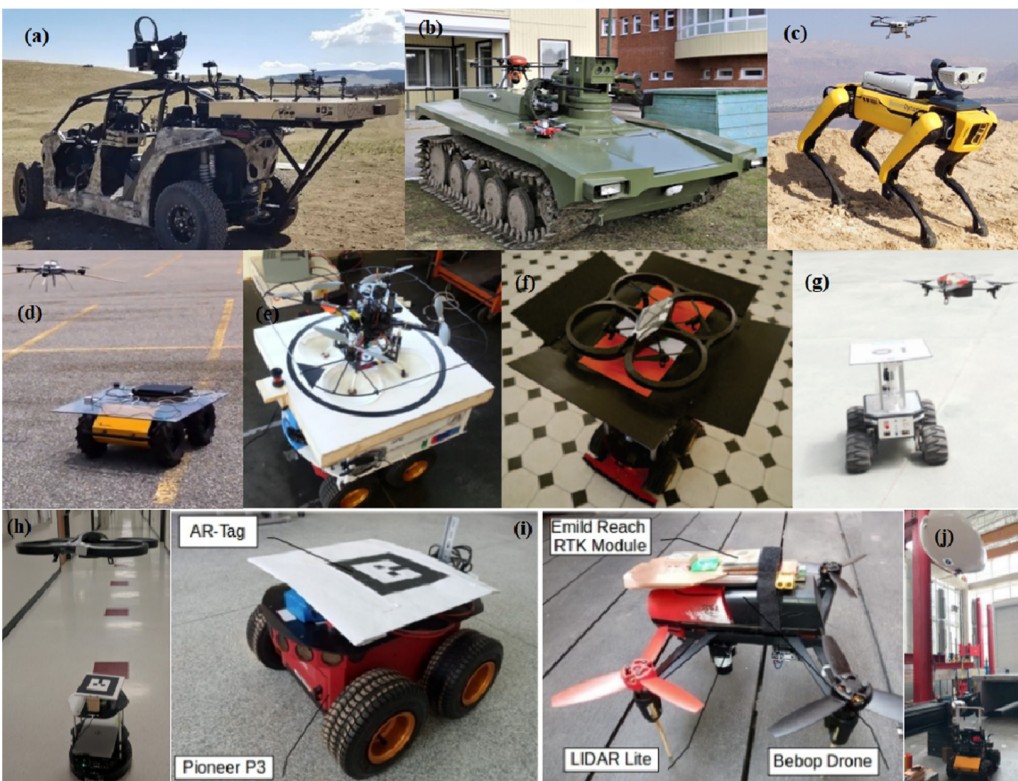

**Figure 45.** Views of drones deployed by UGVs; (**a**) UGV-drone system developed by Planck AeroSystems and Stealth Technologies [82], (**b**) Russian combat robot working with a swarm of drones [83], (**c**) robotic dog/grasshopper from Boston Dynamics carries and deploys a drone [84], (**d**) University of Waterloo's control system that enabled a drone to land safely on a UGV [85], (**e**) UGV-drone interaction developed by Cocchioni et al. [86], (**f**) heterogenous UGV–drone system developed by Saska et al. [87], (**g**) visual-aided autonomous takeoff, tracking, and landing of a drone on a moving UGV by Palafox et al. [88], (**h**) a cooperative exploration using a UGV and a drone proposed by Hood et al. [89], (**i**) a drone and an UGV was examined for detailed inspections of power lines by Cantieri et al. [90], and (**j**) integrated a blimp drone and a Husky UGV tested by Asadi et al. [91].

Saska et al. presented a heterogenous UGV–drone system that could cooperatively solve periodical surveillance tasks in indoor environments (see Figure 45f) [87]. Palafox et al., in 2019, performed the visual-aided autonomous takeoff, tracking, and landing of a drone on a moving UGV (see Figure 45g) [88]. Hood et al. proposed a cooperative ex-

ploration using a UGV and a drone. In this research, the UGV could navigate through the free space, and the drone provided enhanced situational awareness via its higher vantage point. The goal of this application was perform search and rescue in a damaged building (see Figure 45h) [89]. In the paper by Cantieri et al., an architecture involving a drone and an unmanned ground vehicle (UGV) was examined for detailed inspections of power lines (see Figure 45i) [90]. Asadi et al. designed and tested a drone–UGV team that integrated a blimp drone and a Husky UGV (see Figure 45j) for construction environments [91].

### 3.1.1. Tethered Drones Deployed by UGVs

Another application for integrating drones and UGVs is a tethered drone system. This system involves having a drone that is directly tethered to the UGV with cables and wires. A major advantage of the system is that the drone can directly pull power from the UGV's main power supply; this relieves the UGV from carrying the drone-charging system. The tethered drone assists with the UGV's mapping. On the other hand, the major disadvantage of this system is the tether that limits the range of the drone. Another disadvantage of tethering the drone to the UGV is the potential for the tether to become tangled, while the tether also could affect the maneuverability of the UGV. In Figure 46a–c, views of tethered drones deployed by military UGVs are shown [92].

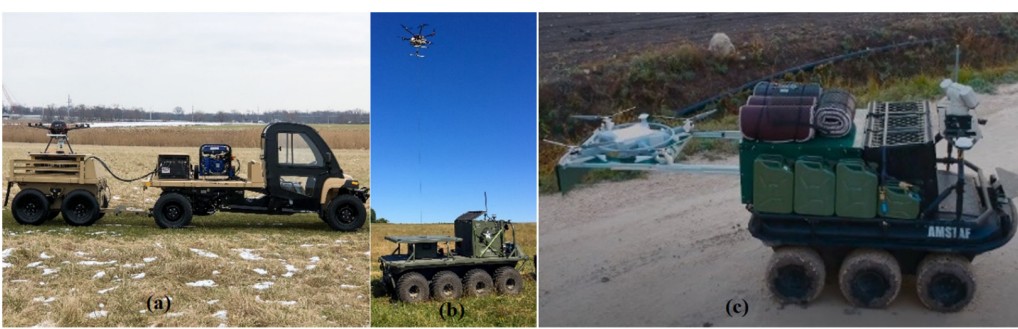

**Figure 46.** Views of tethered drones deployed by military UGVs [92].

In 2014, Papachristos and Tzes worked on autonomous cooperation between a tethered drone and a UGV to improve navigation in partially mapped environments (see Figure 47a) [93]. Miki et al., in 2019, proposed a cooperative system for a drone and a UGV, which uses the drone not only as a flying sensor, but also as a tether attachment device that assists the UGV in climbing a cliff (see Figure 47b) [81]. Nguyen and Garone proposed a system composed of a drone and a UGV, which cooperated to manipulate and move an object to the desired pose [94]. Xiao et al., in 2020, developed an autonomous tethered drone in conjunction with a UGV for operations in unstructured or confined environments (see Figure 47c) [95]. Viegas et al. in 2021, described the design, prototyping, and testing of a novel, lightweight, tethered drone with a mixed multi-rotor design and water jet propulsion for forest fire fighting (see Figure 47d) [96].

In 2022, Martinez-Rozas et al. addressed the problem of trajectory planning in a marsupial robotic system consisting of a drone connected to a UGV through a non-taut tether that has a controllable length. This work aimed to determine a synchronized collision-free trajectory for the three marsupial system agents, including the drone, UGV, and tether [97]. Papachristos et al. presented a complete framework for developing a tethered drone deployed by a ground robot for search and rescue operations (see Figure 47e) [98]. Mullens et al. developed a UGV-mounted automated refueling system for VTOL drones (see Figure 47f) [99]. In 2022, Borgese et al. worked on a multi-robot system composed of a UGV and a tether-connected drone to carry out missions in outdoor environments (see Figure 47g) [100]. Quaglia et al. developed a solar-powered UGV that carries a drone for precision agriculture (see Figure 47h) [101]. Gu et al. proposed a tethered system including roaming tethered aerial robots for radioactive material sampling and stationary tethered

aerial robots for environmental monitoring to meet extremely long endurance missions in nuclear power plants (see Figure 47i) [102].

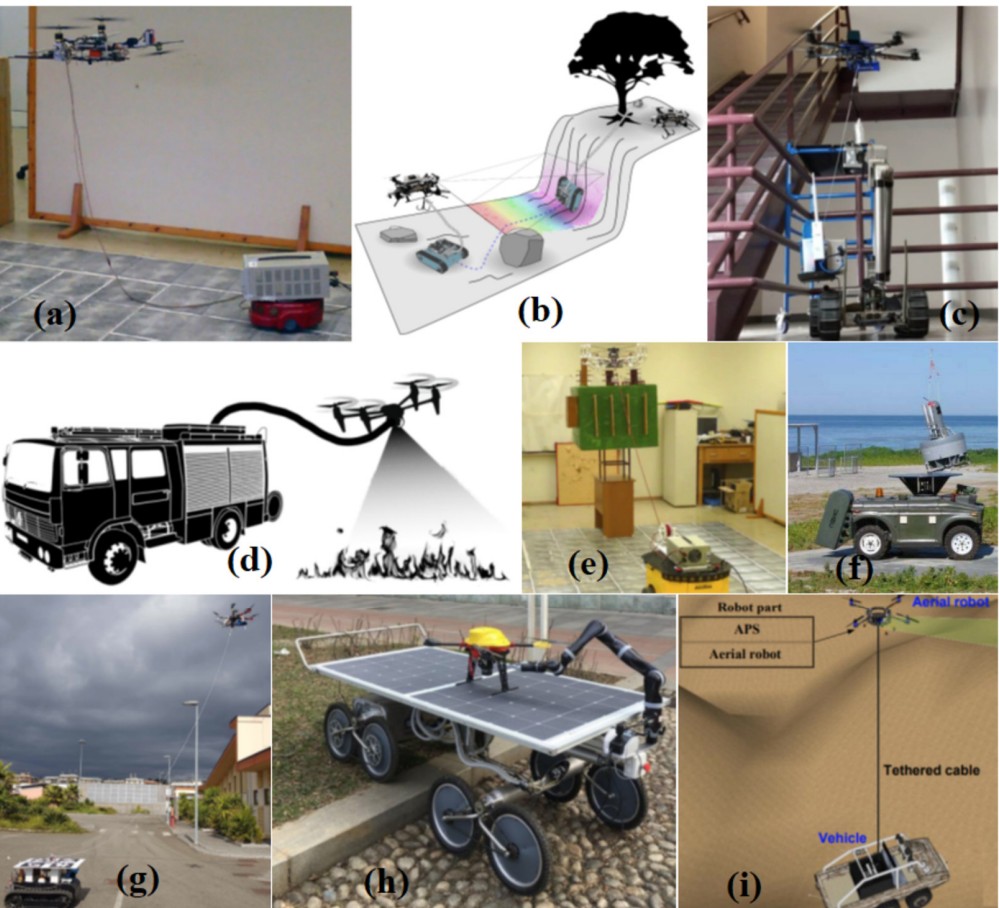

**Figure 47.** Views of tethered drones deployed by UGVs, (**a**) autonomous cooperation between a tethered drone and a UGV developed by Papachristos and Tzes [93], (**b**) cooperative system for a drone and a UGV proposed by Miki et al. [81]., (**c**) an autonomous tethered drone in conjunction with a UGV developed by Xiao et al. [95], (**d**) a tethered drone with a mixed multi-rotor design and water jet propulsion for forest fire fighting designed by Viegas et al. [96], (**e**) a tethered drone deployed by a ground robot for search and rescue operations presented by Papachristos et al. [98], (**f**) UGV-mounted automated refueling system for VTOL drones developed by Mullens et al. [99], (**g**) a UGV and a tether-connected drone designed by Borgese et al. [100], (**h**) a solar-powered UGV that carries a drone for precision agriculture developed by Quaglia et al. [101], and (**i**) a tethered system including roaming tethered aerial robots for radioactive material sampling and stationary tethered aerial robots for environmental monitoring proposed by Gu et al. [102].

### 3.1.2. Housing Platforms for Drones on UGVs

Drone docking stations allow the drones to take off and land and provide a recharging opportunity for them. These stations are generally designed for battery-powered multi-rotor and VTOL drones. For the drones that are deployed by UGVs, the housing mechanism is a key feature of any UGV that houses drones in extreme environments such as underground spaces. An opening and closing hatch is an advantageous type of housing because it is a relatively simple mechanism that protects the housed drone from the elements. The opening and closing hatch mechanisms are also advantageous because they can prevent debris from accumulating on the landing pad while the drone is deployed. Another advantage of the opening and closing hatch mechanism is the potential for opening the hatch to clear debris that may have accumulated on top of the hatch. Another type of housing mechanism is an opening and closing IRIS; IRIS designs are advantageous

because the mechanism does not risk hitting an external object while it is opening. A disadvantage of the IRIS mechanism is the possibility of debris getting stuck on the IRIS while it is closed; this could lead to debris falling on the drone when the IRIS opens for deployment. Figures 48–50 show some of the commercially available housing systems for drones.

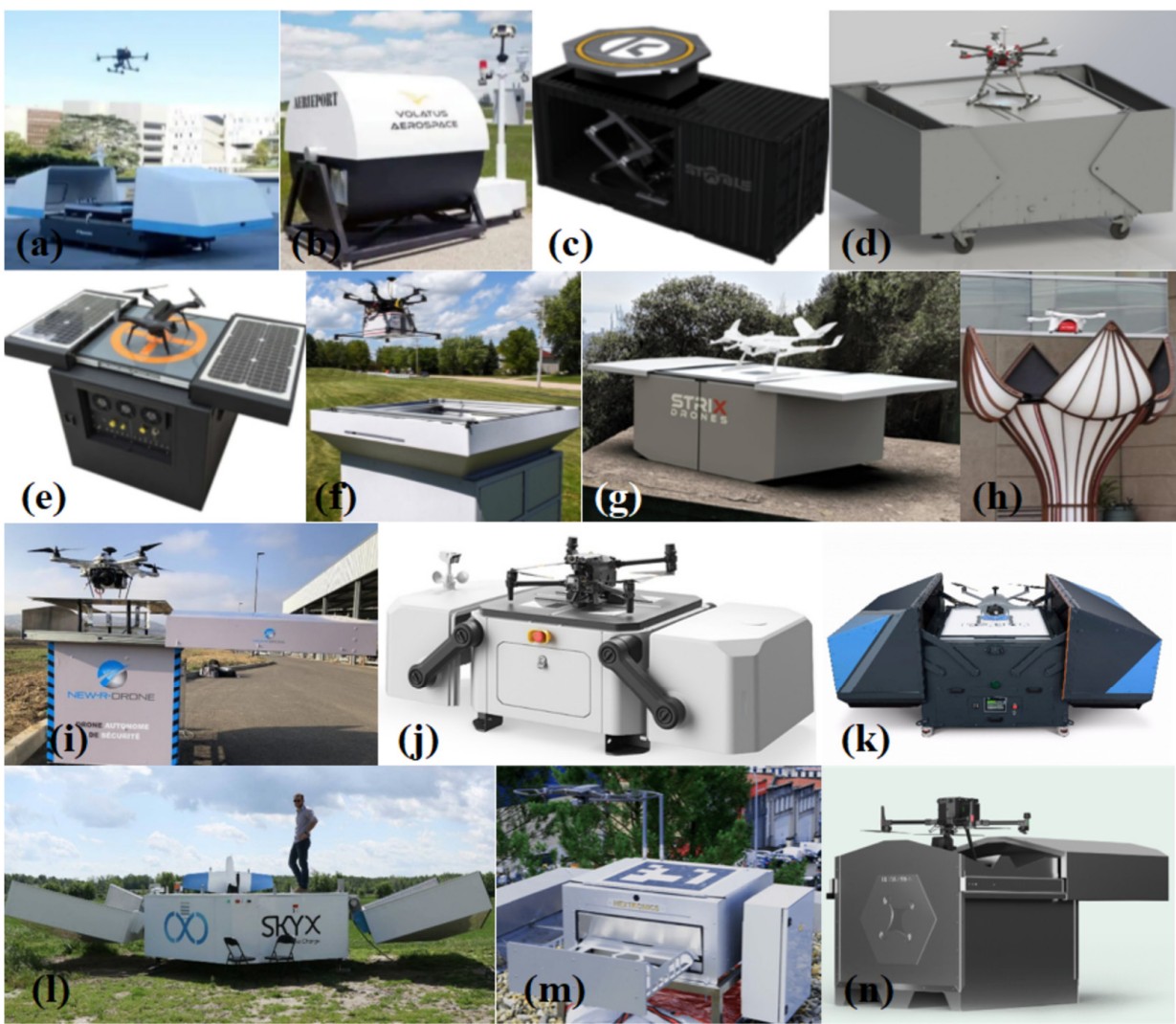

**Figure 48.** Views of (**a**) DBX-G7 drone station by H3 Dynamics [103], (**b**) AERIEPORT by Volatus Aerospace [103], (**c**) stabilized drone landing platform by STABLE [103], (**d**) ENCATA Engineering Catalyst Autonomous drone station [104], (**e**) Singaporean Dronebox, a solar-powered landing station for miniature aircraft [105], (**f**) Valqari's drone delivery station [106], (**g**) Strix Drones, drone agnostic docking stations for advanced operations [107], (**h**) Matternet docking station [108], (**i**) Mobotix drone station [109], (**j**) DJI Dock [110], (**k**) Percepto's Sparrow I drone base station (Courtesy) [111], (**l**) SkyX drone-charging stations [112], (**m**) Hextronics global advanced drone dock [113], and (**n**) Hextronics Atlas 300 docking station [113].

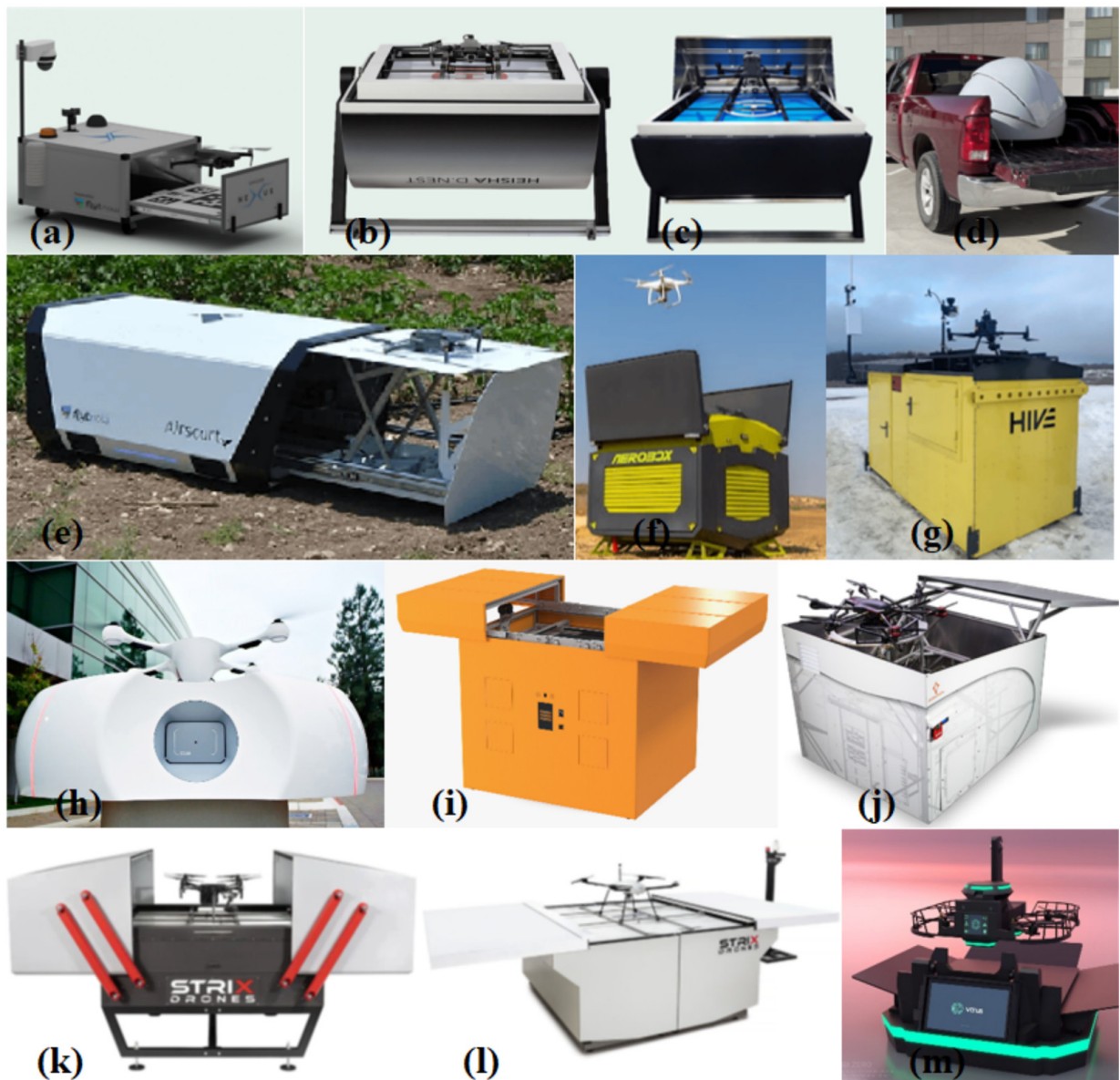

**Figure 49.** Views of (**a**) IDIPLOYER nexus dock [113], (**b**) Heisha D80 dock [113], (**c**) Heisha D135 drone dock [113], (**d**) CounterDome drone station [113], (**e**) Airscort FlytMini [113], (**f**) Aerobox docking station [113], (**g**) Hive Droneport docking station [113], (**h**) Matternet drone station [114], (**i**) 3D Delivery drone station rigged by 3d_molier International [115], (**j**) Airmada automated drone ground stations [116], (**k**) STRIX 1600 drone station [117], (**l**) STRIX 2100 drone station [117], and (**m**) Vtrus ABI Zero drone and base station [118].

As shown in Figures 48–50, these structures house drones in order to protect and charge them. Similar concepts can be designed for the UGV to house and hold the drone before deployment in underground spaces and protect the electrical components from dust. This structure should be fabricated as a box with an opening/closing hatch and shock absorbers to protect and hold the drone during the motion of the UGV.

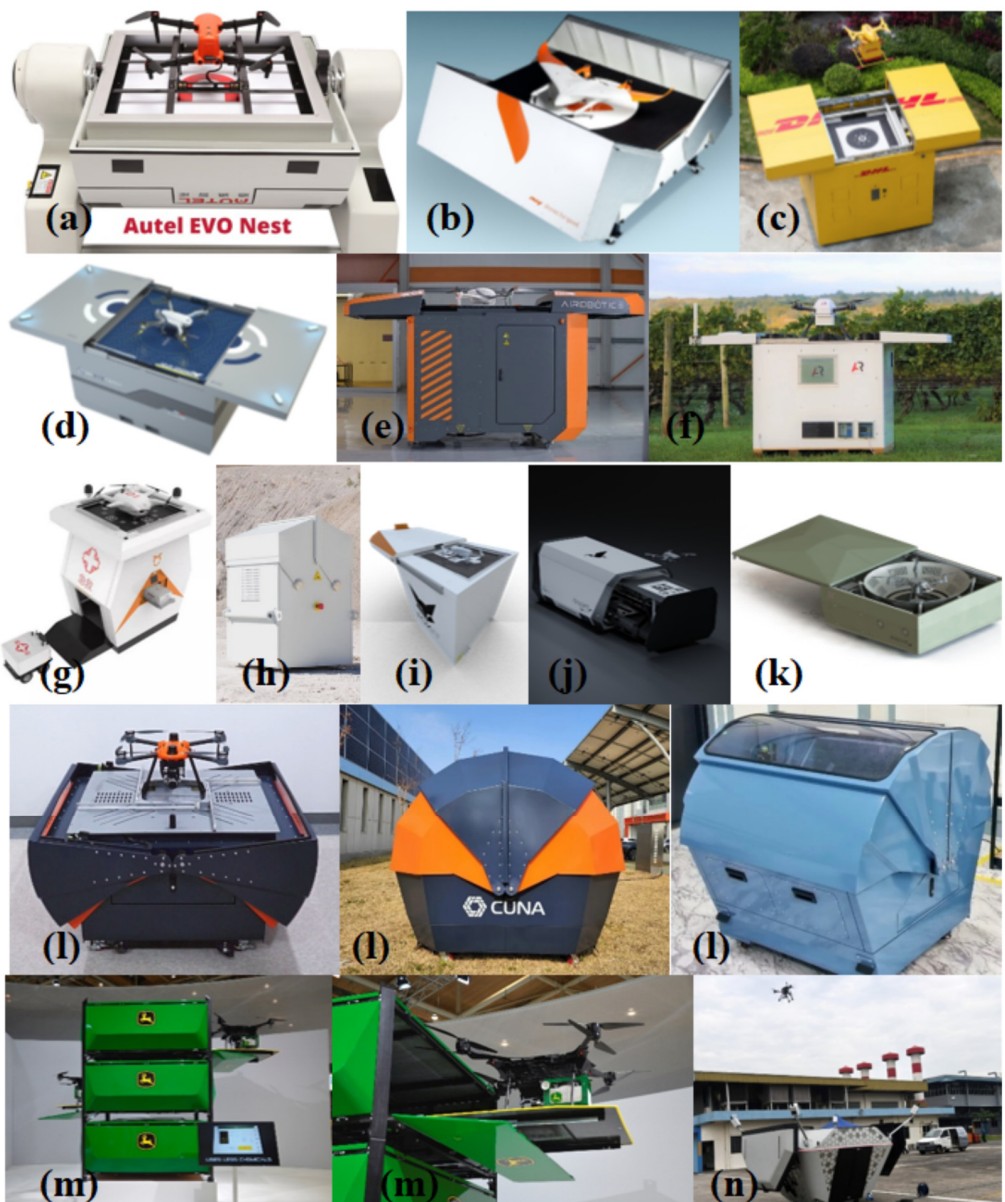

**Figure 50.** Views of (**a**) Autel automatic charging station for EVO II drones [119], (**b**) Avy Dock [120], (**c**) DHL drone dock [121], (**d**) SKEYETECH drone dock [122], (**e**) Airobotics battery-swapping platform [123], (**f**) American Robotics drone station [124], (**g**) Antwork UGV drone dock [125], (**h**) Skycharge Skyport DP5 [126], (**i**) Foxit Response [127], (**j**) Airscort Flytmini [128], (**k**) Airscort [128], (**l**) ARGOSDYNE drone station [129], (**m**) John Deere drone station [130], and (**n**) Singapore Technologies Aerospace [131].

### 3.1.3. Charging Mechanisms for Drones on UGVs

Drone-charging systems can be attached to UGVs to allow them to charge the integrated drones that they house. Drone-charging stations make use of either wired or wireless methods of charging the drone's battery. Wired methods traditionally use a wired contact that can make contact with the drone's battery to charge it or replace the battery. Although wired contact can provide very efficient charging, it requires precise landing, and it poses a danger from the exposed electrical contacts. An autonomous maintenance station should be able to provide a means for guiding drones to land, along with a replacement of the depleted battery with a fully charged battery. The goal of having a battery replacement mechanism in drone docks/housings is to decrease the downtime between the charging of drone batteries in applications, such as smart cities, border patrol, mili-

tary, medical delivery, and postal systems. The specific drones that are typically used are multi-rotors, with typically three or more rotors to produce lift.

Currently, the greatest challenge to overcome with drones is the limited battery capacity and the down time between each charge. There are various methods to keep drones functioning, but some of them are more feasible than others. The proposed solutions include drones outfitted with solar panels, using fossil fuels as the energy source, a combination of solar- and laser-powered drones, and an autonomous battery replacement system. Once, these solutions were evaluated, it is clear the standard limitations of drones become more significant in the design. Other limitations include the total mass, size, maximum altitude, number of propellers, and payload delivery. When one is considering these limitations, it becomes clear that the mass of the drone will most likely make other approaches ineffective, and this leaves only the option of autonomous battery replacement.

One of the first studies was conducted by Fujii et al. [132], where they found success in replacing drone batteries for indoor and outdoor drones. In their design, they illustrated a need for a landing platform, position correction, and battery swapping. Underneath the landing platform was a hidden battery that was preloaded and charged, and once the drone was centered, the battery was then loaded into the drone [132]. In 2015, Nutting and Mercik [133] from Worcester Polytechnic Institute also worked on a similar problem. In this work, they focused more on the locking mechanism of the battery and how it would make contact with the terminals of the drone. They cleverly designed a battery housing with rails on the sides so that the battery could slide into the drone. Their drone and battery case included neodymium magnets as the terminals for powering the drone [133]. As the first two studies were focused on landing stations with battery replacement capabilities, Barrett et al. [134] took a different approach by using a mobile ground base with a robotic arm to replace the batteries of the drones. Alternatively, this design replaced the batteries through the top of the drone, whereas the others slid the batteries into the bottom side. Rather than the landing pad moving the drone, they decided to move the ground base to where the drone landed, and then replace the battery. All of these are viable solutions to keeping the drones in the air [134].

There are a number of ways to approach this problem: To start, the direction from which the battery will be inserted needs to be defined. It is possible that the battery could be inserted from the top, side, or bottom. If the battery is loaded from the top, it ultimately correlates to a robotic arm inserting that battery, which is not the most scalable design. Therefore, it would be ideal to load the battery from either the bottom or the side. Next, a method for moving the drone to the center of the landing pad is required. As discussed by Fujii et al. [132], a landing platform was used with two L-shaped arms with a four-bar linkage (see Figure 51a). With servomotors actuating the four bars, the two L shapes were forced to the center and, respectively, centered the drone. Although this seems to be a good approach, it is missing one key factor: it does not rotate the drone. If a drone is coming in for a landing and some unpredictable event occurs and it lands rotated an arbitrary direction away from the battery swapping mechanism, it could fail to swap out the batteries. Furthermore, this method is not fit for correcting the rotation of the drone. A more effective method has been developed to both center the drone and rotate it to the correct position so that the battery can be removed.

Suzuki et al. [135] used magnetic connections to cause contact between the battery and drone, whereas in the study by Herath et al. [136], they applied passive connections for power and cell balancing. Herath et al. [136], among others, used a simple method of centering and securing the drone, before replacing the battery. The designs included conical cavities for passively guiding the drone into the proper location. The tactic shown below has proven to be effective, but it lacks the compensation for different sized drones (see Figure 51b). Suzuki et al. and Herath et al. [136] used different locking mechanisms for securing the battery, but both of the groups loaded the battery into the drone from the bottom. A rack and pinion mechanism was used to push the battery onto a platform

that elevated, inserting the battery into a mechanical coupler. Alternatively, the method proposed allowed the battery to slide horizontally through the drone [136,137].

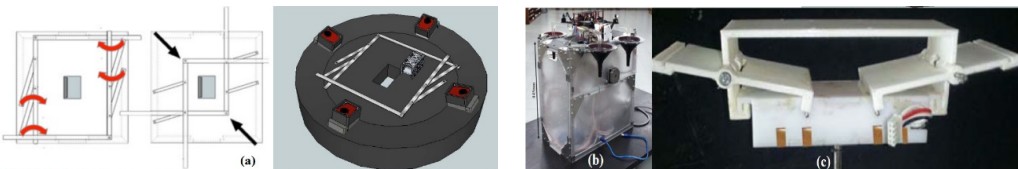

**Figure 51.** View of (**a**) self-centering landing platform for drone [132], (**b**) passive landing guides, and (**c**) passive connectors for power and cell balancing [136].

Wireless charging provides an alternative to wired charging. Depending on the type of wireless charger, oriented landings may or may not be necessary [138]. A slight disadvantage of wireless charging is that the drone/UGV will have to be equipped with receiver coils to be able to charge (see Figure 52).

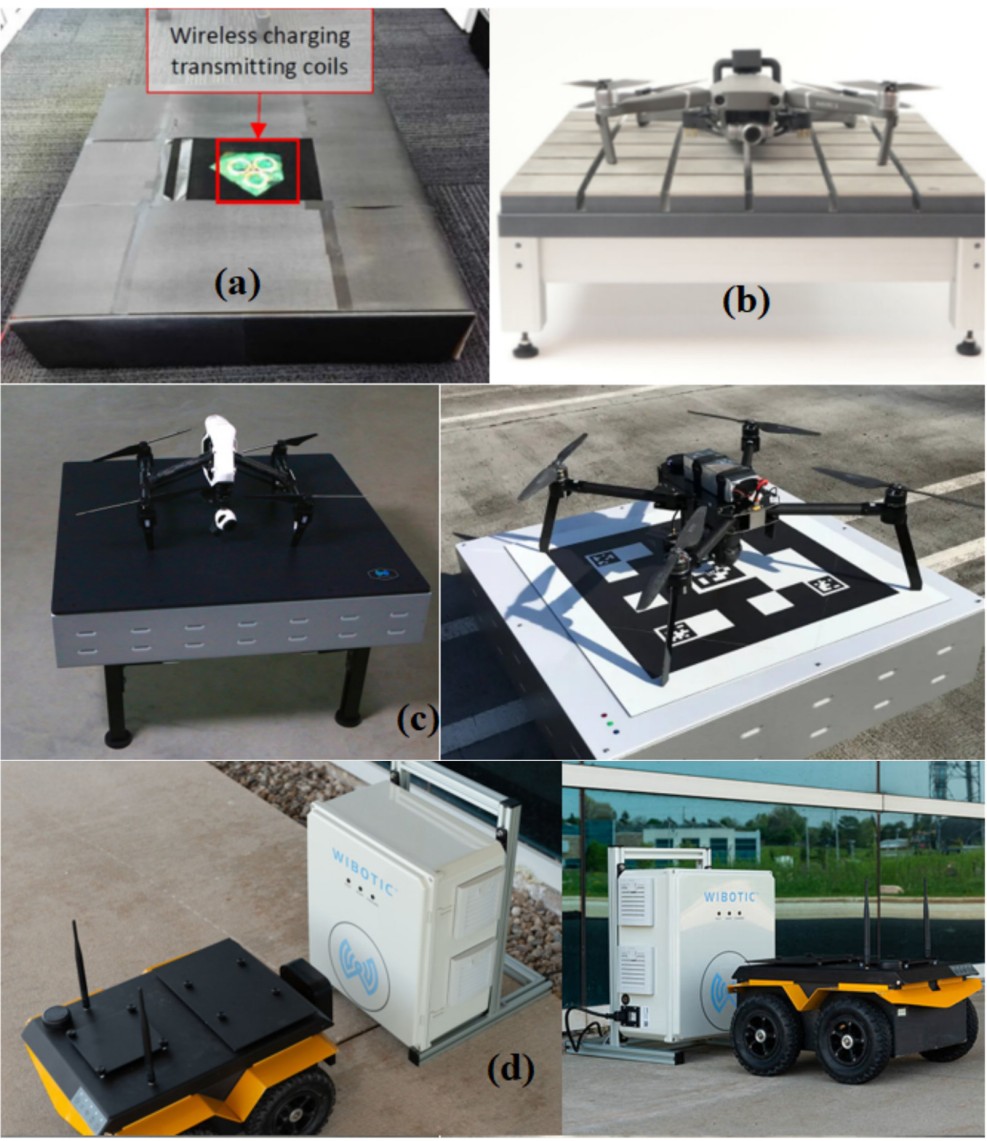

**Figure 52.** View of (**a**) a landing pad that uses wireless charging coils to charge quadcopters with receiver coils [139], (**b**) SKYCHARGE drone wireless charging pad [140], (**c**) WiBotic drone-charging pad [141], and (**d**) WiBotic UGV charging pad [141].

### 3.2. UGVs Deployed by Drones

UGVs can be deployed by drones in an integrated manner. Having a UGV be deployed by a drone is advantageous for several reasons. One of the main advantages is that the drone can fly the UGV over terrain that would have been impossible to navigate by the UGV. Another advantage is that the drone can also be used to fly the UGV directly to and from the mission area, which can save time and shorten the UGV's driving time. One potential disadvantage is that the UGV's size and weight are limited by the drone's carrying capacity.

Galeone developed an autonomous drone that is capable of transporting a ground robot through a gripping system (see Figure 53a) [142]. Hament and Oh developed a technical design requirements for a team of autonomous drones and UGVs for civil infrastructure inspection (see Figure 53b) [143]. In 2022, the Heven Drones Company designed and built a heavy lift drone that can carry a Micro Tactical Ground Robot (MTGR) (see Figure 53c) [144]. In 2022, a group in China tested a drone deploying a quadrupedal unmanned ground vehicle (see Figure 53d) [145]. The Korea Atomic Energy Research Institute used a combination of a drone and a snake robot for accident monitoring (see Figure 53e) [146]. Kawasaki's K-Racer tested an autonomous helicopter, which could carry a cargo robot (see Figure 53f) [147]. Nagatani et al. developed a robotic system for volcano exploration using drones transporting ground robots (see Figure 53g) [148].

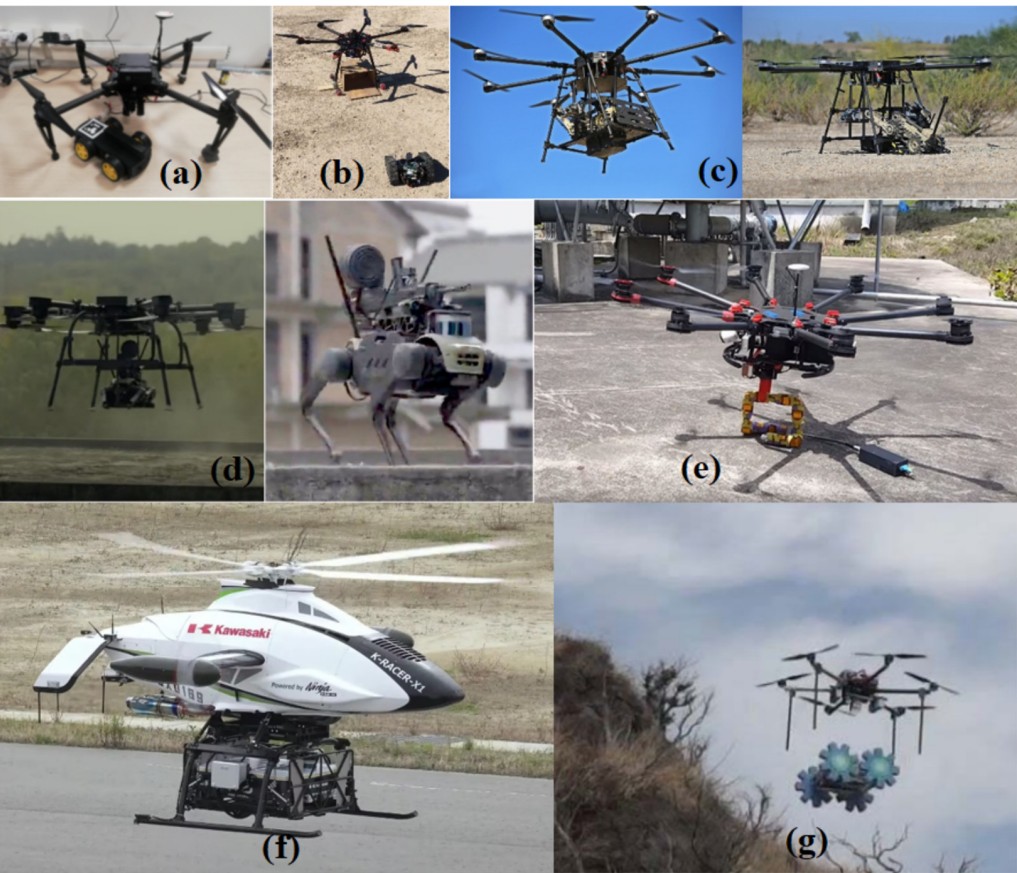

**Figure 53.** View of (**a**) drone transporting a ground robot [142], (**b**) team of autonomous drones and UGVs for civil infrastructure inspection [143], (**c**) a drone that can carry a Micro Tactical Ground Robot (MTGR) [144], (**d**) drone deploying a quadrupedal unmanned ground vehicle [145], (**e**) a drone that can carry a snake robot for accident monitoring [146], (**f**) autonomous helicopter which transport a cargo robot [147], and (**g**) a robotic systems for volcano exploration [148].

## 4. Mining Challenges for Multi-Agent Systems

The environmental conditions in an underground mine present some challenges to the humans and machinery that work there, making their deployment difficult, if not impossible altogether. Studies into the use of autonomous systems have made it obvious that the proper utilization of these systems can significantly improve the search and rescue outcomes of underground mining emergencies. The intent for their deployment is generally to hasten the rescue mission process because, in an emergency, time is of the essence. They also help to reach inaccessible parts of the mine relatively easier and identify trapped miners quicker. It is no doubt that these systems are going to be faced with performance-impeding conditions [4]. Addressed in this section are the challenges the robots are going to face.

### 4.1. Darkness

Since the inception of underground mining activities, illumination has been pivotal to the safety and performance of both miners and machinery. Every country has specifications for lighting intensity that are used in their mining industry (see Table 1).

**Table 1.** Illumination levels around machinery in the mining industries in different countries [149].

| Country | Illuminance (Lux) |
|---|---|
| Belgium | 25 |
| Hungry | 20–30 |
| Canada (British Columbia) | 53 |
| Poland | 10 |
| Germany | 80 |
| United States of America | 0.061 Fl/021 (cd/m$^2$) |

The United States is the only country that opted for the measure of luminance over illuminance, as it is more correlated to what the eye perceives [150]. The lighting conditions in an underground mine, and especially, in an underground coal mine, are affected by a number of factors including (but not limited to) the following:

Poorly reflective walls: Reflectivity is a measure of the ability of a surface to reflect the incident light supplied to it. In lighting applications, surfaces or walls reflect a relatively high percentage of light, which helps in providing good visibility of the objects. In underground coal mines, exposed surfaces, and walls, especially around the ribs and faces, have very low reflectance, usually 1–10%, with 4% being the typical value [150]. Coupled with the low reflective nature of the environment is the texture-less and locally similar structure [151]. This presents localizability issues and severe sensor degradation. To address this issue, Mascarich et al. designed a multi-modal unit that incorporates a visible light camera with inertial sensors, as well as bright LEDs that flash only when the shutter opens [152]. When the multi-modal unit is used together with a visual-inertial odometry pipeline, it enables reliable robot navigation in the darkness. Additionally, other researchers have developed a system that integrates the robot's multi-modal system with a set of associated fusion algorithms that enable Simultaneous Localization and Mapping (SLAM) and the mapping of visually degraded environments in an underground mine [151].

Strict regulations surrounding the use of electrical equipment: Provision 75.1719–1(d) of the Codes of Federal Regulations (Mine Safety & Health Administration, 1988 [153]) states that: "The luminous intensity (surface brightness) of surfaces that are in a miner's normal field of vision of areas in working places that are required to be lighted shall not be less than 0.2 cd/m2 (0.6fl)." Additionally, for underground mines, the voltage of the lighting system should not exceed 125 volts. The electrical equipment must be intrinsically safe, which means it has to be safe for use in atmospheres containing flammable gasses and combustible dust [154]. Therefore, it implies that the electrical device should be incapable of producing enough thermal or electrical energy to ignite the fuel and cause an explosion. These regulations are in place to mitigate accidents and disasters, and rightly so,

underground mines make it impossible to use high luminescence output lamps in underground mines. For this reason, more research is required to know the feasibility of using high-output or low-voltage lighting systems.

### 4.2. Dust and Smoke

Underground mines can be broadly categorized into two main types: non-coal and coal ones. The non-coal mines are generally more metalliferous and deep-seated ones relative to the coal mines. Underground non-coal mines are considered to be relatively less dangerous than coal mines are because there are generally no poisonous emissions in those mines [155]. Underground coal mines are surfeited with noxious gasses such as carbon monoxide and firedamp (methane–air mixture), which can explode with the application of the slightest amount of heat and produce thick clouds of dust and smoke. The occurrence of spontaneous combustion in coal mines and the presence of dust and smoke can render the robots ineffective by directly impeding their capacity to self-localize and locate trapped miners with any accuracy [156].

Underground mine exploration activities produce heavy dust and smoke that obscures distinctive features and landmarks that the system may rely on for localization. According to [156], visual-inertial SLAM methods are unstable and can lose track or even enter unrecoverable states in such environments. The research on overcoming this issue is widely ongoing [157]. Reliance on LiDAR sensing modalities is the best available option, though it is insufficient. However, recent research focuses on multi-modal sensor fusion and robust state estimations to provide reliable alternatives [158].

### 4.3. GPS-Denied Environements

The implementation of UGV and drones is becoming an increasingly efficient tool for rescue applications, site and infrastructure inspection, remote controlling, exploration, path finding, and risk recognition [18,25,29,35,38]. During an underground emergency, the location of the base points plays a fundamental role in the rescue mission, and the hybrid robotic system can help to locate trapped miners easily. Ideally, the UGV and drones would be equipped with a Global Positioning System (GPS) to enable it to self-localize and map out the underground mine; however, the depth of the mine below the ground level, the rock composition, and the harsh environment in an underground mine make it difficult to maintain any usable GPS signal [4].

To mitigate this problem, robots deployed for GPS-deprived applications implement alternative positioning algorithms, such as Simultaneous Localization and Mapping (SLAM), Partially Observable Markov Decision Process (POMDP), Dijkstra, or Model Predictive Control (MPC), to mention a few [4–6]. These algorithms can help the robots to self-localize by using measurements from features inside the mine to track the occluded targets in GPS-denied atmospheres and map the morphed underground tunnels. The parameters in these algorithms will depend on the location, motion, and speed of the unmanned vehicle performing in the underground environment [4–6,151].

According to MSHA, these base points or targets are the keys to streamlining the rescues, finding the miners in an emergency, and keeping the rescue team safe [153]. For that reason, the GPS-denied environment is one of the most crucial challenges the robot can face in underground mines. [159] The location of strategic points, such as a fresh air base, allows the rescue teams to communicate between the command center and plan strategies [159,160].

Another strategic point is the refuge chamber: it isolates the miners from the hazardous event and has tools for establishing communication with the command center and rescue team to notify the situation in the mine in real time [161,162].

### 4.4. Dynamic and Harsh Environment

In the last two decades, improvements to the regulations and technological applications have helped to minimize accidents in underground mines. The fatalities and financial losses have decreased considerably compared with those in the last century [155,159]; however, the inherently harsh and dynamic nature of the underground environment continues to present the mining industry with a lot of challenges [157]. The harsh environment refers to the factors that potentially lead to a degraded perception of miners, difficulty breathing, inaccessibility to certain areas, and unstable communication signals in rescue practices and mining work. Some of these factors are dust, smoke, darkness, toxic gasses, unpassable obstacles, temperature, and humidity. These conditions are characterized by constant and sometimes sudden changes due to mining activity or progressing through the ore body, making an underground mine a dynamic atmosphere [154].

In 2006 in Sago Mine (coal mine), eleven out of twelve miners died of carbon monoxide poisoning before the rescuers reached them 41 h after a disaster [154]. The Sago Mine incident exemplifies the dynamic spaces and harsh environment that the miners and rescue teams must consider during incidents [154,159]. The occurrence of an explosion could cause the roof to collapse, and this can produce dense dust and debris. The debris or rubble can end up blocking some spaces, creating blind spots; this makes the exploration process complex [158]. These obstructed areas serve as toxic gas concentration sites because air cannot circulate through them. Additionally, the presence of gasses such as methane and oxygen can cause another explosion, and the cycle may be repeated. These conditions tend to slow down the already slow process of mine rescue, as the safety of the rescue team can also be endangered if these factors are ignored. The use of the UGV–drone hybrid mitigates the risk that rescuers would otherwise face in the mine because they can traverse toxic environments, reach obstructed spaces, and see in the darkness. This helps to significantly accelerate the rescue protocols and procedures [154,155,162].

The technology integrated into the UGV and drone monitors accurate results of the conditions under challenging environments and determines the occurring events in the mine. Technology such as RGB-depth (RGB-D) cameras measure the distance between obstacles. The hybrid UGV–drone system can reach more search areas; when the UGV cannot advance due to an obstruction, the drone will continue the exploration by air [162]. Similarly, the thermal camera can identify fire and analyze the robot's risk conditions during rescue procedures.

### 4.5. Ventilation and Air Velocity

Mine ventilation is the process of supplying sufficient fresh air to the underground environment through a ventilation system and working to ensure its proper distribution and use using controls that return contaminated air to the surface [163]. This fresh air volume must be sufficient to dilute and remove dust and noxious gasses, typically $SO_2$, $CH_4$, $CO_2$, and $CO$, and regulate the temperature inside the mine [155,159,164]. The ventilation system is closely related to the development of work and rescue procedures. Confined spaces do not allow efficient air to flow through the mine; this makes the underground mine a place where harmful gasses and pollutants easily concentrate, with the least amount of deficiency in the ventilation system [158,165]. Fresh air enters the mine from the surface via the hoisting shafts and is distributed strategically using a series of fans and the space circulation (raises and ramps) of the mine to maintain the airflow [159,166].

The most important design parameter in monitoring the functionality of the ventilation system is air velocity; this parameter plays an important role in health and safety hazard management [159,164]. The ventilation system may have a stable flow, allowing the person to identify critical locations in real time; the air velocity can be measured using different sensors or techniques such as a rotating vane anemometer, a velometer airflow measure probe, a piezoresistive sensor, and an ultrasonic flowmeter [165,167]. During a disaster, however, the situation may be quite different. Fires or explosions may release dangerous gasses into the atmosphere. A disrupted ventilation system could result in an

oxygen-deficient atmosphere and/or a buildup of toxic or explosive gasses [154,159]. The flow's velocity and direction permit the identification of critical locations where the gasses are concentrated at and measures them using technology integrated into the UGV and drone, such as thermal cameras, LiDAR, sensors, and others.

According to Mine Rescue Team Training in metal and non-metal mines, gas detection is essential to any rescue or recovery operation. The rescue teams must perform frequent tests for gasses as it advances beyond the fresh air base, identifying what harmful gasses are present, how much oxygen is in the atmosphere, and whether the gas levels are within the explosive range [152,159]. The sensors integrated into the UGV–drone system could monitor the gas concentrations, such as methane from 0.0 to 100 percent of volume, oxygen from 0.0 percent to at least 20 percent of volume, and carbon monoxide from 0.0 parts per million to at least 9,999 parts per million (30 CFR 49.6(a)(6)) [164].

The Fresh Air Base (FAB) is a structure designed as a temporary stop during an emergency by the miners and the rescue team [159]. The base is used to plan, rest, move forward and execute the evacuation/rescue operations [168]. It is equipped with a supply of potable water and compressed air, isolating the toxic gasses and giving rescue teams a breath of fresh air. Additionally, the base usually has a copy of the most recent emergency procedures, the map of the mine, and the communication route to the surface [167]. The ventilation engineers design the fresh air base based on the fresh air source, and it is connected to the ventilation system to guarantee breathable air across the mine. The appropriate use of the base permits them to establish communication between the rescue team and the UGV–drone system, increasing the communication range, guaranteeing the control of the robot, and making the signal more stable [169,170].

The rescue personnel can control the drones in the fresh air base, increasing the coverage of the UGV and drone, protecting the rescue team from hazards present in the disaster, and providing fresh air or breathing apparatus with oxygen. The control of a robot in a safe place reduces the risk posed to the rescuers and speeds up the rescue procedures. "A team's time underground is usually limited to two hours or less. The exact amount of time is determined both by the underground conditions and the type of apparatus being used. The distance you can advance also depends on underground conditions." [158,159].

### 4.6. Muddy Terrain

The rescue and self-escape missions are heavily time-dependent missions. The faster that these missions are executed, the higher the survival chance of the trapped miners is. One of the reasons that UGVs and drones have been deployed is to reduce the mission's execution time. The floor of the gallery is typically harsh and uneven with protuberances and even cracks on its surface. This condition does not hinder the performance of the machinery underground because they have large tires or tracks. In the event of an emergency, such as a roof fall, these roadways become filled with piles of rubble. Additionally, in the case of an inundation, heavy dust settling in water can form a thick slurry or muddy waters. The unevenness of the roadway, the access paths being filled with rubble, and the presence of muddy terrain can impede the movement of the robots [151]. Sandia National Laboratory, recognizing this challenge, developed the "Gemini Scout'', shown in Figure 54, which overcame these movement-restricting factors [171].

The Gemini scout UGV is equipped with gas sensors, thermal cameras, and a tall, mounted tilt camera. At the same time, it is lightweight enough to crawl over boulders, rubble piles, and standing or flowing water. Its articulated body and rubber tracks enable it to maneuver over rough terrains.

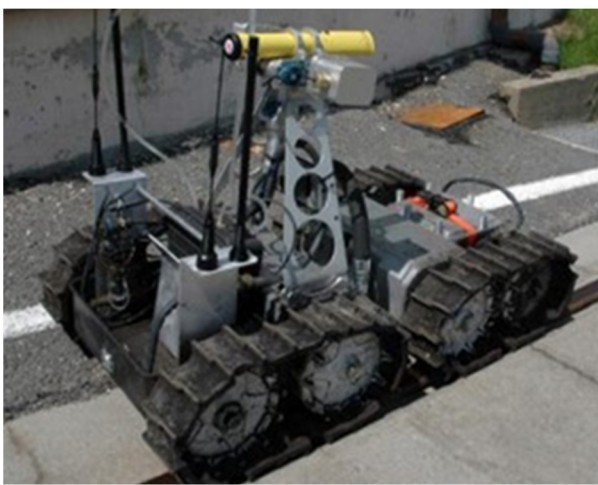

**Figure 54.** View of Gemini Scout UGV [171].

*4.7. Permissibility*

Permissibility refers to the consent by the Mine Safety and Health Administration (MSHA) to allow electric face equipment used for mining operations, including mining rescue, in the mine [37]. The certification of approval is one of the barriers to the electronic function of the UGV and the drone, especially in coal mines. Permissibility is a major barrier to implementing such technologies.

According to MSHA (2012) in the 30 CFR § 75.506, "Electric face equipment", requirements for permissibility related to "electric-driven mine equipment and accessories manufactured on or after 30 March 1973, will be permissible electric face equipment only (1) if they are fabricated, assembled, or built under an approval, or any extension thereof, issued by the Bureau of Mines or the Mine Safety and Health Administration in accordance with schedule 2G, or any subsequent Bureau of Mines schedule promulgated by the Secretary after 30 March 1970, which amends, modifies, or supersedes the permissibility requirements of schedule 2G, and (2) if they are maintained in a permissible condition." [27,37].

The MSHA is the regulatory body in charge of granting permissibility certifications and practicing tests to guarantee the health and safety of mines [155]. The high concentrations of gasses amidst electrical equipment can generate an explosion due to the ignition of electrical devices incorporated in the robot [154,159,170]. However, the application of underground mining technologies, at least in the case of the hybrid ones, helps with the enhancement of rescue procedures, minimizing the manpower, and restoring reliability in mining operations. Therefore, the industry continues to investigate the application of permissive gadget technology in subway mines to maintain the safety of miners and rescue teams [27,154]. In addition, the multi-agent system implemented through UGVs and drones maintains the permissive requirements in terms of software design. The software has the capability to become an intrinsically safe system. Human factors and inadequate procedures are major contributors to fatalities in emergency events, and the area of autonomy and navigation for mine emergency applications will significantly improve health and safety.

*4.8. Wireless Communication*

A critical aspect of a multi-agent hybrid robotic system is the efficient and secure communication between the robotic agents necessary for transmitting the data. Similarly to the infeasibility of deploying GPS technology in subterranean environments, significant challenges arise for all forms of wired or wireless communication due to the lack of the infrastructure or the impracticality of installing such infrastructure necessary for communicating the information-carrying signal. Therefore, the design and deployment of an MAHRS must address this lack of communication tools by incorporating appropriate technologies

and methodologies that enable a capability inherent to MAHRS to create and manage its own communication network.

Vehicle-to-Vehicle (V2V) communication technologies could provide the means to achieve efficient inter-communication between the robotic agents of MAHRSs. V2V communication has attracted the interest of considerable research in the field of Intelligent Transportation Systems (ITS) in recent years. Although, V2V is mainly studied in ITS applications, with the main purpose of enhancing road safety and driver protection (through integrating the speed and orientation of one vehicle with the collision zones of the surrounding vehicles), the same technologies can be applied for enabling efficient data communication between the members of MAHRSs. V2V is primarily achieved by employing Vehicular Ad Hoc Networks (VANETs) between vehicles, i.e., spontaneously created wireless networks of vehicles, in order to exchange data between two moving autonomous vehicles. These data can include collected sensor data, vehicle size, position, speed, heading, lateral/longitudinal acceleration, yaw rate, throttle position, brake status, steering angle, wiper status, turn signal status, and loss of stability information.

VANETs can be built by using any wireless networking technology. The most common ones are short range radio technologies such as WLAN, and especially, the IEEE 802.11p WLAN protocol, which was built with the intention to be used for vehicular networks. Cellular networks are also a promising candidate for the basis of making a VANET. Infrared transmission and reception can also be used for creating these wireless networks by using Visible Light Communication [172].

Araniti et al. examined the capabilities of using LTE for supporting V2V communications, as well as the advantages of LTE over the 802.11p [173]. They conclude that the strengths of LTE (high capacity, wide coverage, and high penetration) can cope with the drawbacks of 802.11p (poor scalability, low capacity, and intermittent connectivity). Visible Light Communication (VLC) based on LEDs for transmitting data is another applicable methodology. In environments were autonomous agents have to operate under smoke, dust, or poor illumination conditions, which deteriorate the onboard sensors accuracy and increase the risk of collisions, LED-based V2V technologies could offer promising solutions [174].

Besides VANETs, flying ad hoc networks (FANETs) have attracted a lot of interest in the last two decades. Drones, due to their high maneuverability and scalability, have been the most popular candidates to create autonomous ad hoc networks [175–177]. As a result, there is an abundance of literature sources that rigorously investigate multiple aspects of designing and implementing swarms of drones, which include (but are not limited to): swarm coordination [178–180], routing [181], security [182–187], and energy considerations [188], as well as their challenges and prospective uses [189].

*4.9. Power Sources: Energy for Multi-Agent Systems*

In underground environments such as coal mines, fuel- and electrically powered machines pose two principal problems: they are a potential fire and explosion hazard and produce noxious exhaust gasses that must be diluted to safe levels. Manufacturers of these machines incorporate special design features that aim to produce the least amount of heat and noxious gasses. The MSHA approval and certification center evaluates the power package to ensure that the special design features provide the protection that is required, and if federal requirements have been met, a certification is issued. Concerning the use of multi-agent systems underground, all forms of power supplied to this system and any electrical systems installed on it must incorporate permissibility protection provided by the certified power package.

4.9.1. Batteries

The most important feature of a UGV is mobility. The power supply's duration is undoubtedly one of the most important indicators for evaluating a UGV, aside from the working temperature range. According to MSHA, battery-powered equipment should have an

intrinsically safe active voltage circuit, which is a circuit that utilizes non-passive components such as transistors or other solid-state devices that either shunt overvoltage or overcurrent conditions or limit the output of the power source fast enough to prevent the ignition of a methane–air atmosphere [190]. The most common power supply source for UGVs is a battery, especially for small- and medium-sized UGVs. Batteries have some advantages over combustion engines: they are quiet, mechanically simple, and, most importantly, have lower working temperature ranges and no heated exhaust. This is preferable, especially for use in environments with methane–air concentrations.

Battery Type and Permissibility

UGVs, drones, and robotic platforms are commonly powered by batteries. The weight, size, and endurance of the battery are limitations of the performance of many autonomous vehicles [191]. In underground rescue applications, unmanned systems need a reliable backup power system to operate during prolonged recovery procedures, and it is a continuous challenge to select the best batteries that will perform safely [192]. The voltage level, capacity, discharge rate, activation time, charging time, lifespan, and their cost efficiency are some of the parameters that must be considered when one is selecting a battery pack for unmanaged systems. Autonomous vehicles have an integrated technology that operates with different functions. That means the battery loads and power demands vary depending on the operation requirements carried out in underground mines.

Lithium-based batteries are the most used technology in unmanned systems due to their higher energy density and charge–recharge efficiency [193]. Mostly, the unmanaged system industry uses lithium polymer (Li-Po) and lithium–ion (Li-ion) batteries to develop complete systems in autonomous vehicles. Furthermore, they are developing technology using lithium sulfur batteries (Li-S), where the energy density is even higher compared with that of other Lithium-based batteries [194]. Additionally, the old technology implemented in unmanaged systems was nickel-base batteries, but their usage has decreased in frequency due to the power-to-weight ratios for these types of batteries in drone engines not being efficient for the rate of climb and power consumption [195].

According to the National Institute for Occupational Safety and Health (NIOSH), the uses of Li ion batteries in some devices employed in mining are permissible for communication and tracking devices. Nevertheless, the Li ion battery cells pose a CH4 explosion hazard from a cell internal short circuit. The use of this type of battery requires further testing to rule out its use and permissibility in underground [194] mines. The sealed enclosure for batteries is one of the keys means of avoiding the ignition of explosions or fires in underground mines. On the other hand, an MSHA-approved Li ion cap lamp battery contains a multicell battery pack within a sealed plastic enclosure [192]. The batteries used lithium ion technology, which requires special consideration and handling techniques due to the extremely high energy density [196]. Quality control tests for permissible Li ion and Li polymer batteries are carried out to standardize their use in coal mining equipment, including UGVs and drones [196].

4.9.2. Fuel Gas

Secured operations of diesel-powered machinery in underground coal mines, where methane and other flammable gasses may be present, require custom features to mitigate fire and explosions. These features are listed in part 36, Title 30 of the Code of Federal Regulations (30 CFR 36). A permissibility checklist has been developed to recognize the diverse nature of the techniques used by equipment manufacturers to provide the required protection and the importance of ensuring that the safe devices continue to provide the required protection. If a diesel-powered engine system is going to be incorporated into a multi-agent system, then the engine and exhaust system surfaces must be maintained below 302⊚ [197]. One of the most effective means for reliably controlling this temperature has proven to be water jacketing the exhaust manifold and the associated piping. If the temperature around the engine goes beyond the above-mentioned limit, combustible

concentrations of methane surrounding the engine may be ignited by flames passing back through the engine intake system, passing through the joints in the engine intake or exhaust system.

### 4.9.3. Liquid Hydrogen

Liquid hydrogen is one of the most common fuels used in fuel cells. The supply of hydrogen is incomparable to that of conventional energy sources such as diesel, gasoline, or electricity due to their production scale. Additionally, hydrogen storage is the most serious limiting factor to its effectiveness and distribution. Despite the fact that hydrogen is as clean as the energy that is used to produce it is, liquid hydrogen in UGVs must be compressed in expensive high-pressure tanks, which requires more energy. Regardless of all of the disadvantages of using hydrogen fuel, hydrogen cell systems can outperform battery systems by a factor of 2–8 (within the 100–1500 Watt power range) [198]. This translates to a longer performance time and a better performance of the electrical components.

## 5. Application of Hybrid Systems in Underground Spaces

Multiple researchers have focused on integrating different sensors into a navigation system that is capable of exploring and collecting data in subterranean or similar GPS-deprived spaces, such as catacombs, caves, and mines.

Although there are innumerable sources on the international literature showcasing the development of UGVs deployed in a GPS-denied environment, both surface and underground applications, an exhaustive review is omitted for the sake of simplicity. However, some examples of autonomous UGVs deployed in the mining industry can be insightful regarding the vast possible capabilities of such vehicles: Lee et al. deployed the Cave Crawler platform equipped with a spinning laser scanner (i.e., LiDAR), an IMU (Inertial Measurement Unit), and wheel encoders in an underground coal mine (see Figure 55a). A band-based ICP algorithm (Iterative Closest Point) was developed for reconstructing a narrow band from the 3D point cloud of the mine tunnels to reduce the computational cost [199]. Nagatani et al. integrated a 2D LiDAR scanner with an infrared camera with a thermal ICP algorithm to develop 3D thermography maps, which help to locate humans (see Figure 55b) [200]. Lavigne and Marshall integrated a laser scanner and an IMU with an RFID reader (Radio Frequency Identification) that detects passive RFID beacons for a landmark-based method for mitigating drift in large-scale underground mine mapping (see Figure 55c) [201]. Neumann et al. implemented a SLAM algorithm using ROS packages (Robotic Operating System) to fuse the data from LiDARs, IMU, and wheel encoders into a collision avoidance navigation framework for underground mines (see Figure 55d) [202].

Similarly, there is an abundance of sources in the international literature that showcase the development and the capabilities of drones in a plethora of environments [3,4,91,203,204]. The deployment of drones in mines or similar GPS-deprived spaces has attracted a fair share of research interest, especially in recent years, which have advanced localization and mapping methodologies (e.g., SLAM), as well as computational power and efficiency. Shahmoradi et al. provided a comprehensive review of the deployment of drones in the mining industry, with applications that range from 3D mapping and geotechnical characterization to mine operation monitoring, air quality monitoring, subsidence monitoring, and mine rescue missions [4]. Jones et al. showcased a multitude of similar implementations of drone technology in underground mining operations, focusing on the Hovermap drone system that was developed by the Australian Commonwealth Scientific and Industrial Research Organization (CSIRO) for GPS-deprived environments [205].

The recent advances in computing, sensors, hardware, and software have led to an increasing interest in deploying cooperative UGV–UAV systems in GPS-denied environments. Asadi et al. showed that researchers have been exploring the capabilities of multi-agent UGV–UAV systems since the 1990s for different environments and applications. The reviewed applications are collectively summarized in a tabular format in a very insightful

way [91]. However, the same study concludes that most of the reviewed applications do not operate the collaborative robots simultaneously or autonomously. More relevant research on the deployment of multi-agent systems in underground, GPS-deprived spaces was conducted through the 'DARPA Subterranean Challenge' (Defense Advanced Research Projects Agency). This challenge brought together researchers and innovators from around the world to develop systems that can map, navigate, and execute time-critical tasks in a complex subterranean environment that combines tunnel systems (i.e., underground mines), urban underground layouts, and natural cave networks. The vast variety of the multi-agent systems that competed in the challenge included (but were not limited to): the winning team CERBERUS, who combined walking and flying robots [206], the team CSIRO Data61, who deployed multiple types of ground and air robots: a hexapod, a quadruped robot, a UAV, and three tracked robots [207], the CTU-CRAS-NORLAB team, who deployed wheeled, tracked and crawling (hexapod) ground robots and quadrotor aerial robots [208], while the NeBula team combined wheeled, tracked, quadruped, aerial (quadrotor and coax-octocopter), as well as hybrid robots [209].

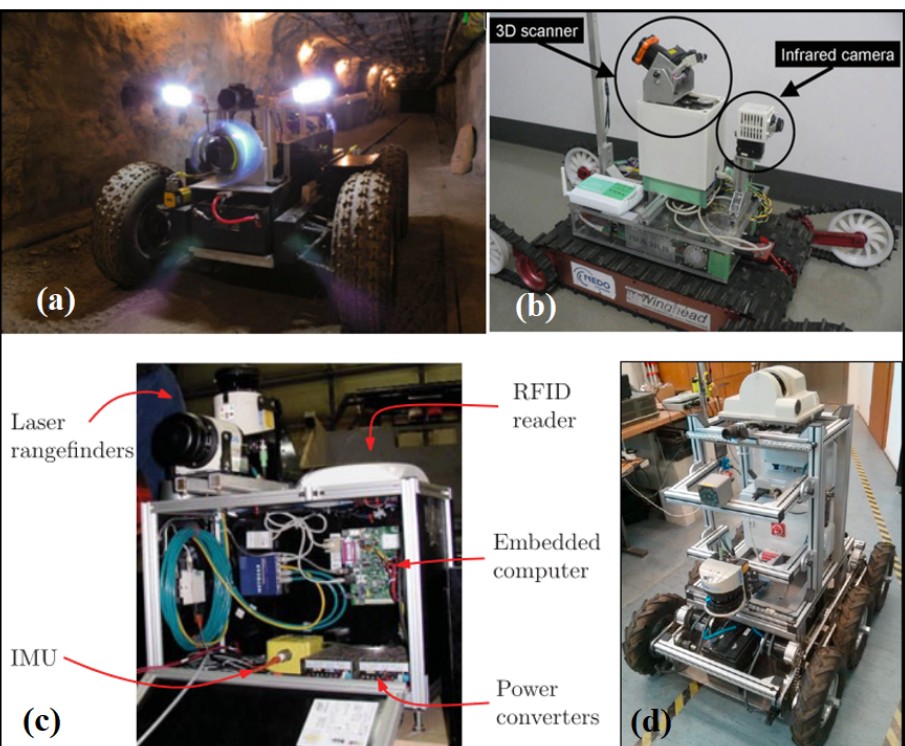

**Figure 55.** View of (**a**) the Cave Crawler platform with a spinning laser scanner in an underground coal mine [199], (**b**) Kenaf robot with a 3D laser scanner and an infrared camera [200], (**c**) the custom hardware enclosure of the landmark-based mapping platform with a laser scanner, an IMU, and a RFID reader [201], and (**d**) the collision avoidance mobile testing platform [202].

## 6. Conclusions

This review focuses primarily on unmanned ground and aerial vehicles, either as stand-alone agents or as multi-agent systems (hybrid) in underground spaces. The categories of unmanned surface and underwater vehicles (USVs and UUVs), which encompass autonomous agents that operate on the surface and under the surface of the water, respectively, are not discussed in this review. The principal reason for this omission is that these types of unmanned vehicles have little or no applicability in underground environments or can be easily replaced by hybrid UGV–UAV systems. In addition to the examination of the various configurations of ground vehicles and drones prevalent in the global literature, the category of bio-inspired robots is briefly examined. The emphasis of the examination

of these robotic agents is on their efficiency and capacity to carry out, either as stand-alone agents or as part of an MAHRS, conducting exploratory and surveying missions in subterranean spaces.

The literature shows an abundance of different configurations of ground vehicles (UGVs) and drones (UAVs), as well as coupled or inter-transformable UGV–UAV systems that have the capacity for autonomous navigation, and hence, the exploration of subterranean spaces. A similar abundance of cooperative combinations of these unmanned vehicles can be found in the literature. These multi-agent hybrid robotic systems (MAHRSs) are meant to be as terrain versatile as possible by using specific unmanned robots to support and make up for the shortcomings of the other cooperative agents. A major advantage of MAHRSs is the efficiency, in terms of speed and deployment time, that they can contribute in a mission. Such systems assign separate subtasks to each of their robotic agents according to their built-in capabilities. In this way, the ultimate mission of the system is executed more speedily and efficiently.

The principal aspects that a robot's efficiency is commonly measured against include (but are not restricted to) the following: size, energy capacity/consumption, and movement control. Each of these parameters determines some capabilities, as well as the effectiveness of them. At the same time, these parameters are not independent of each other, but instead they affect and are affected by each other. Although a detailed discussion of these aspects is out of the scope of this study, the following indicative observations can be made. Size and weight are important considerations for any hybrid system that will be used within an underground mine. UGVs that deploy drones will both have increased sizes and weights compared to those of a UGV deployed via a drone because they must carry the apparatus to deploy and house the drones. The size of a robot determines the feasible deployment working environments, the overload capacity, and its maneuverability. The overload capacity of a robot determines whether it can house and deploy other agents or can be deployed by other agents. Moreover, the overload capacity directly determines the onboard hardware, and hence, the computational power and overall functionality of the robot. Naturally, the size of a robot affects its battery consumption and maneuverability. The energy capacity/consumption relates directly to the deployment time, onboard hardware, computational cost, and actuation mechanisms. In other words, heavier and smarter robots usually require more energy, but they can accomplish more complex tasks. In general, the sizes of underground mines necessitate that any deployed system should have a long endurance capacity. UGVs that deploy tethered or untethered drones will consume a significant amount of power due to their large size and the need to power or charge the deployed drones; however, they can compensate for this by carrying a large battery that can give them a long endurance capacity. A drone that deploys a UGV would consume a lot of power because it has to fly the UGV to its destination; this would reduce the endurance of the system. The deployed UGV would also have a limited battery size and weight because it has to be light enough to be picked up by the drone; this would limit its endurance once it has been deployed. Finally, movement control determines the feasible deployment working environment of the robot, whereas it affects energy consumption and is interrelated to the size. The complexity that arises from the interconnections and codependency of such design aspects is evident even when we are considering a few basic aspects. Therefore, the reasons underlying the popularity of designing MAHRS becomes obvious.

Hybrid systems consisting of unmanned aerial and ground vehicles can efficiently explore and navigate underground spaces such as catacombs, caves, and mines. The cooperative implementation of drones and ground-traversing vehicles means that they have a greater exploration capacity in dynamic underground environments that cannot be easily accessed from the ground. Naturally formed underground cavities such as caverns and lava tubes are usually inaccessible or even inhospitable for wheeled vehicles. Similarly, man-made underground openings, such as mines or tunnels, can be restrictive to rovers by construction or be rendered as such due to caving, infrastructure failures, or erosion.

Exploring such underground spaces with a hybrid system means that multiple traversing alternatives can be crucial to exploration missions. Spaces which cannot be accessed by rovers or humans may be explorable by small, precise drones that can fly through tight spaces or reach high places to gather environmental data. These data can aid mine rescue teams or be used for surveying new environments.

Depending on the environment, UGVs deployed by drones could have the quickest deployment time because they can fly quickly to the deployment area, whereas UGVs that deploy either tethered or non-tethered drones must drive to their destination, which could take longer. Rovers deployed by drones are at a major disadvantage in underground mines because of the difficulty of flying in confined spaces while carrying the significant and cumbersome load of the rover. On the other side, tethered and non-tethered drones deployed by UGVs exhibit advantages in accessing underground mines because the UGVs can drive them to their deployment area. Subsequently, the deployed drones can utilize their relatively smaller size, which enables flight in confined spaces. Tethered drones deployed by UGVs can have a slight disadvantage depending on the intended range of the drone and the number of obstacles in the mine for the tether to overcome.

Bio-inspired robots can also be integrated in an MAHRS. The capabilities and efficiency of an MAHRS can be expanded or improved by such robots. Although, most applications in the literature showcase bio-inspired robots that are small in size, and hence, do not have the capacity to support advanced functionalities (with the exception of the legged robots such as the ones shown in Figure 10). Nevertheless, such small robots can be the ideal candidates for critical subtasks in the successful deployment of an MAHRS. Assigning the roles of wireless network nodes, distinctive landmarks, or signal beacons, or data stations to such robots would render vast flexibility to an MAHRS.

In any case, the robotic agents of an MAHRS that intend to be used for deployment in underground mines need to be carefully designed in order to overcome the challenges imposed by that particular environment: the poor illumination, as well as the presence of dust and smoke, impose limitations to the applicability of sensor modalities depending on the light conditions, while they can significantly occludes the data collected by other sensors; the GPS-denied nature of the subterranean spaces does not allow the use of one of the most powerful localization technologies; the dynamic and feature-less nature of the mines adds more complexity to autonomous navigation processes; ventilation and air velocity fluctuations create disturbances in the movement of UAVs, while uneven and muddy terrain can hinder the movement of a UGV; finally, permissibility regulations have to be considered in the design and fabrication of the robots. Nevertheless, multiple researchers have developed single, hybrid, or multi-agent systems that are optimized in some aspect for mines or similar subterranean environments, with insightful results.

To conclude, the ultimate scope of this study is to provide a collective literature review by drawing insights from Mineral Engineering R&D and Intelligent Robotic Systems R&D that aims to improve and expedite future R&D on intelligent underground systems in an effort to facilitate and inspire designers and innovators of such systems.

**Author Contributions:** Conceptualization, M.H., P.R., V.A., H.K. and S.S.; methodology, C.D. (Chris Dinelli), J.R., M.E., S.L., J.G., J.M., C.D. (Chase Dunaway) and M.H.; validation, V.A., H.K., S.S., P.R. and M.H.; formal analysis, C.D. (Chris Dinelli), J.R. and M.E.; investigation, C.D. (Chris Dinelli), J.R., M.E., V.A., H.K., S.S., P.R. and M.H.; resources, M.H. and P.R.; writing—original draft preparation, C.D. (Chris Dinelli), J.R., M.E., S.L., J.G., J.M., C.D. (Chase Dunaway) and M.H.; writing—review and editing, V.A., H.K., S.S., P.R. and M.H.; visualization, C.D. (Chris Dinelli), J.R. and M.E.; supervision, M.H. and P.R.; project administration, S.S., P.R. and M.H.; funding acquisition, M.H., P.R. and S.S. All authors have read and agreed to the published version of the manuscript.

**Funding:** This research was funded by NIOSH-CDC: grant number U60 OH012351.

**Institutional Review Board Statement:** Not applicable.

**Informed Consent Statement:** Not applicable.

**Data Availability Statement:** All data are available in the article.

**Acknowledgments:** This study was funded by the National Institute for Occupational Safety and Health (NIOSH) under the award #U60OH012351. The views, opinions, and recommendations expressed herein are solely those of the authors and do not necessarily reflect the views of NIOSH. Mentions of trade names, commercial products, or organizations does not imply endorsement by the authors nor the funding organization.

**Conflicts of Interest:** The authors declare no conflict of interest.

## Abbreviations

| Acronyms | Definitions |
| --- | --- |
| 2D | Two-Dimensional |
| 3D | Three-Dimensional |
| CFR | Code of Federal Regulations |
| CSIRO | (Australian) Commonwealth Scientific and Industrial Research Organization |
| DARPA | Defense Advanced Research Projects Agency |
| DOF | Degrees of Freedom |
| FAB | Fresh Air Base |
| FANET | Flying Ad hoc Network |
| GPS | Global Positioning System |
| ICP | Iterative Closest Point |
| IMU | Inertial Measurement Unit |
| ITS | Intelligent Transportation System |
| LED | Light Emitting Diode |
| LEMUR | Limbed Excursion Mechanical Utility Robot |
| LiDAR | Light Detection and Ranging |
| Li-ion | Lithium ion |
| Li-Po | Lithium Polymer |
| Li-S | Lithium–Sulfur |
| LTE | Long-Term Evolution |
| MAHRS | Multiple-Agent Hybrid Robotic System |
| MPC | Model Predictive Control |
| MTGR | Micro Tactical Ground Robot |
| MMC | Moving-Mass Controlled |
| MSHA | Mine Safety and Health Administration |
| POMDP | Partially Observable Markov Decision Process |
| R&D | Research and Development |
| RFID | Radio Frequency Identification |
| RGB | Red–Green–Blue |
| RGB-D | Red–Green–Blue–Depth |
| ROS | Robotic Operating System |
| SLAM | Simultaneous Localization and Mapping |
| SMA | Shape Memory Alloy |
| UAV | Unmanned Aerial Vehicle |
| UGV | Unmanned Ground Vehicle |
| USV | Unmanned Surface Vehicle |
| UUV | Unmanned Underwater Vehicle |
| V2V | Vehicle-to-Vehicle |
| VANET | Vehicular Ad Hoc Network |
| VTOL | Vertical Takeoff and Landing |
| VLC | Visible Light Communication |
| WLAN | Wireless Local Area Network |

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
