# Peer review of "Configurations and Applications of Multi-Agent Hybrid Drone/Unmanned Ground Vehicle for Underground Environments: A Review"

_drones, doi:10.3390/drones7020136_

Round 1

Reviewer 1 Report

1-      ome titles like sections 2.1, 2.1.3, 2.2.5, 2.3, … are just mentioned in headers without any body. It is recommended to provide an introduction for these sections at least summarizing the coming sub-sections.

2-      It would be helpful if you provide illustrations for sections 2 and 3 summarizing the classifications that are studied in these two sections.

3-      It would be a good idea if you provide more detailed underground applications in section 5 and review the missions which have been done so far using hybrid robotic systems.

Author Response

Please find attached our responses.

Reviewer 2 Report

The key contributions as claimed by the authors are:

"This review presents a collective study of robotic systems, both of individual and hybrid types, intended for deployment in such environments. The prevalent practices for the construction and hardware equipment, as well as the artificial intelligence of existing multi-agent hybrid robotic systems, will be discussed. This review aims to provide a supplementary tool for defining the state-of-the-art of coupled Unmanned Ground Vehicle (UGV)-drone systems implemented for underground exploration and navigation purposes, as well as to provide some suggestions on multi-agent drone system solutions and ultimately to support the development of a semi-autonomous multi-agent hybrid vehicle to assist mine emergency response.

It is comprehensive study which is based on good reference literature on unmanned systems and Cooperative UGV-Drone Systems. The English writing and organization of this article is fine. There are some minor issues such as.

1: The quality of some figures is not good. The authors must provide figures with high resolution quality.

2: Line 17, what is GPS? The authors are suggested to carefully check each acronym and must define at the first place. 

3: In Introduction, authors can incorporate a Table and compare their work with some existing reviews or surveys. 

4. Section 5 does not contain any sufficient discussion and references. Either it should be removed and adjusted in other section or it must be revised with sufficient reference literature.

5. Conclusion section must be revised and it must be more precise.

6. Several references are out date and many references are taken from websites. It is suggested to provide relevant academic references such as Journals or Conferences (If possible).

7. Authors can include motivation, scope and organization of this review as well. 

Overall, it is interesting article which contains good discussion. It will be very helpful for relevant research fraternity, 

Author Response

Please find attached our responses. 

Reviewer 3 Report

- The article covers an interesting topic and will serve as a good reference for researchers and scientists interested in "Multi-Agent Hybrid 2 Drone-Unmanned Ground Vehicles" (in the event of its acceptance and publication, of course).

  - The paper is extremely long, and I think it is necessary to include: 1. A table of Content  2. A table of Figures  3. A table of Tables  4. A index of keywords and corresponding page numbers.   - At the end of the introduction, the authors need to add a short paragraph that describes the structure of the paper.   - In addition, the authors may add a new figure that illustrates the structure of the paper.   - The authors are also invited to add a new paragraph in the introduction which explains their search methodology and priorities (keywords, databases, inclusion/exclusion criteria, etc.)   - It is also necessary to add a new related work section that covers similar surveys related to the same topic for emphasizing the originality of this paper.   -  The authors are also invited to add a new paragraph in the introduction or the related work section about the security aspects related to UAV communications.   - For this purpose, they are invited to consider/insert the following interesting reference (and others): 1.  https://ieeexplore.ieee.org/document/9842403 2. https://link.springer.com/chapter/10.1007/978-3-319-94496-8_7  3. https://doi.org/10.3390/drones6010010 - Economical and environmental aspects need to be studied too.   - Since the paper contains a lot of information and text, the authors need to summarize the content of each section in a tabular format to facilitate understanding.   - Limitations of existing solutions, along with the current challenges and future possible evolution in the considered field need to be presented as well.   - Section 5 is extremely short compared to other sections.    - Since the number of authors is huge (12 authors), it is necessary to identify the contribution of each author .        

Author Response

Please find attached our responses. 

Round 2

Reviewer 3 Report

Some of my comments were not taken into account in the right way and to the right extent. The authors, in particular, overlooked a critical point concerning the security of drone communications.  For this purpose, please consider the previously suggested references.

In addition, the authors are invited to report on the use of well-known formal methods for checking and guaranteeing the correctness of IoT-based systems.

For this purpose they may consider the following references (and others):

- https://link.springer.com/article/10.1007/s11036-019-01369-6

- https://link.springer.com/chapter/10.1007/978-3-642-24270-0_17
